# Directional Gradient Projection for Robust Fine-Tuning of Foundation Models

**Chengyue Huang, Junjiao Tian, Brisa Maneechotesuwan, Shivang Chopra, Zsolt Kira**
Georgia Institute of Technology
`{chuang475,jtian73,bmaneech3,shivangchopra11,zkira}@gatech.edu`

## Abstract

Robust fine-tuning aims to adapt large foundation models to downstream tasks while preserving their robustness to distribution shifts. Existing methods primarily focus on constraining and projecting current model towards the pre-trained initialization based on the magnitudes between fine-tuned and pre-trained weights, which often require extensive hyper-parameter tuning and can sometimes result in underfitting. In this work, we propose **Di**rectional **Gra**dient **P**rojection (*DiGraP*), a novel layer-wise trainable method that incorporates directional information from gradients to bridge regularization and multi-objective optimization. Besides demonstrating our method on image classification, as another contribution we generalize this area to the multi-modal evaluation settings for robust fine-tuning. Specifically, we first bridge the uni-modal and multi-modal gap by performing analysis on Image Classification reformulated Visual Question Answering (VQA) benchmarks and further categorize ten out-of-distribution (OOD) VQA datasets by distribution shift types and degree (i.e. near versus far OOD). Experimental results show that *DiGraP* consistently outperforms existing baselines across Image Classfication and VQA tasks with discriminative and generative backbones, improving both in-distribution (ID) generalization and OOD robustness.[1]

## 1 Introduction

Robust fine-tuning has become an essential technique in adapting pre-trained models to downstream tasks, particularly in the face of distribution shifts that challenge model generalization. While pre-trained models excel in capturing a wide range of features from diverse datasets, fine-tuning them on specific tasks often leads to overfitting, reducing their robustness to out-of-distribution (OOD) data (Wortsman et al., 2022; Nguyen et al., 2024). The goal of robust fine-tuning is to strike a balance between task-specific performance and maintaining the generalization abilities of the pre-trained model (Wortsman et al., 2022). This is particularly crucial for real-world applications such as visual question answering (VQA), where models are frequently exposed to varying distributions in images, questions, and answers (Agrawal et al., 2018; Shah et al., 2019). Effective robust fine-tuning strategies aim to mitigate performance degradation by incorporating techniques like regularization (Li et al., 2018) and bi-level optimization (Tian et al., 2023a;b; 2024), ensuring that models retain their learned knowledge while adapting to new domains.

To tailor the model for downstream tasks while retaining the capabilities of the pre-trained model (e.g. robustness to distribution shifts), L2-SP (Li et al., 2018) imposes a regularization term on the distance between the fine-tuned and pre-trained weights. More recently, instead of viewing robust fine-tuning as a regularization problem, TPGM (Tian et al., 2023a) and FTP (Tian et al., 2023b) consider the regularization term as the constraint to reformulate the problem from a *bi-level optimization* prospective and propose to learn different hard constraints for each layer. However, these methods are computationally heavy and often apply overly strong constraints which results in underfitting (See Sec. 4.2 and Sec. 4.3). Moreover, these methods perform weight projection to enforce the distance between fine-tuned and pre-trained weights within a set of projection radii, which is magnitude-wise but does not encode any directional information. *This motivates us to think of direction-based methods for this fundamental problem.*

---

[1]The code is available at `https://github.com/chengyuehuang511/DiGraP`

We re-examine the regularization problem from a multi-objective optimization perspective and propose **Di**rectional **Gra**dient **P**rojection (*DiGraP*). We consider the regularization term as the second objective besides the first objective, i.e., original loss function, and the goal is to minimize the two simultaneously. This new method involves two aspects: the *projecting condition* and the *directional projection strength*. If the projecting condition meets, i.e., the gradient directions of the two objectives are opposite, the decrease of one objective will lead to the increase of the other objective. In this case, we project the opposite gradient to the orthogonal direction of the other to get rid of the conflicting directional information. We also add a directional projection strength $\omega \in [0, 1]$ to learn the priority of the two objectives and make it trainable so that it can be dynamic throughout the training process.

- Regularization (L2-SP): $\min \mathcal{L}(\theta) = \tilde{\mathcal{L}}(\theta) + \frac{\lambda}{2} \|\theta - \theta_0\|_2^2$
- Bi-level optimization (TPGM, FTP): $\min \tilde{\mathcal{L}}(\theta)$ s.t. $\|\theta - \theta_0\|_2^2 \leq \gamma$
- Multi-objective optimization (*DiGraP*): $\min\{\tilde{\mathcal{L}}(\theta), \frac{1}{2}\|\theta - \theta_0\|_2^2\}$

Typical evaluations on robust fine-tuning methods are primarily done in uni-modal settings, e.g., image classification or semantic segmentation. However, they have not been analyzed in the context of multiple modalities. We first bridge the gap by comparing our method with baselines on an Image Classification reformulated VQA benchmark. We further propose a new setting for evaluating robust fine-tuning of VQA by leveraging ten VQA datasets and categorizing them into in-distribution (ID), near and far OOD datasets covering uni-modal, multi-modal and adversarial distribution shifts. *DiGraP* achieves SOTA ID and OOD performance on both image classification and VQA benchmarks. Our contributions are:

- We propose a layer-wise trainable directional gradient projection method *DiGraP* for robust fine-tuning of large foundation models with the intuition of bridging regularization and multi-objective optimization. This is the first robust fine-tuning methods that considers the directional information of the gradients.
- We propose new settings for evaluating robust fine-tuning of VQA. We first conduct experiments on Image Classification reformulated VQA benchmarks. We then categorize the existing OOD VQA datasets into different types and degrees of distribution shifts and present a consistent comparative analysis of robust fine-tuning algorithms.
- We show that *DiGraP* consistently outperforms other baselines across uni-modal and multi-modal tasks in both ID generelization and OOD robustness.

## 2   RELATED WORKS

**Robust Fine-Tuning of Foundation Models.** LP-FT (Kumar et al., 2022) proposes a two-step strategy of linear probing then full fine-tuning to prevent the feature distortion of the pre-trained layers. WiSE-FT (Wortsman et al., 2022) interpolates the pre-trained and fine-tuned weights to combine the strengths of the two embedding space. L2-SP (Li et al., 2018) explicitly adds an L2 norm penalty on the deviation between the fine-tuned and pre-trained weights. MARS-SP (Gouk et al., 2021) further studies different forms of norms as the penalty. More recently, TPGM (Tian et al., 2023a) approaches the regularization term as a constraint, reformulating the problem as a *bi-level optimization* and proposing to learn distinct hard constraints for each layer. FTP (Tian et al., 2023b) further improves the efficiency of TPGM (Tian et al., 2023a) by learning the constraint from training set of previous step instead of the current validation set. We argue that these methods are still computationally heavy and requires lots of tuning, whereas *DiGraP* reformulates the problems as *multi-objective optimization*, injects directional information and is more intuitive to tune.

**OOD Robustness in VQA.** Previous works have proposed various settings for evaluating robust VQA models. Agrawal et al. (2023) conducts cross-dataset evaluations with four VQA datasets, while Ma et al. (2024) and Li et al. (2021) provides a more comprehensive and detailed robustness analysis by incorporating VQAv2 variants and categorizing different types of distribution shifts. We build on these efforts by introducing further granularity with near and far OOD distinctions and measuring the distance between distributions. More importantly, while previous work has primarily focused on testing different backbone models, they have not yet compared different robust fine-tuning methods.

# 3   DIRECTIONAL GRADIENT PROJECTION FOR ROBUST FINE-TUNING

In this section, we first describe the intuition and the mathematical motivation behind *DiGraP*. Then, we provide our method's concrete algorithmic design.

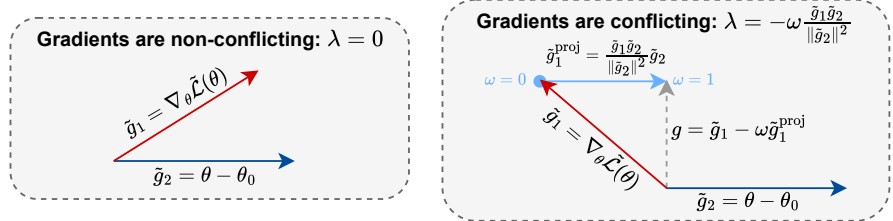

Figure 1: **Directional Gradient Projection.** (Left) When the gradients are non-conflicting, we do not perform projection. (Right) When the gradients are conflicting, we project the gradient for the original loss function according to the gradient for the regularization term. We add $\omega \in [0, 1]$ to control the projection strength, where $\omega = 0$ is an unconstrained update and $\omega = 1$ represents a full projection to the orthogonal direction of the gradient for the regularization term.

## 3.1   ROBUST FINE-TUNING AS A MULTI-OBJECTIVE OPTIMIZATION PROBLEM

In order to adapt the model to the downstream tasks but preserve the power of the pre-trained model (e.g. robustness to distribution shifts), methods such as L2-SP (Li et al., 2018) impose a regularization term on the distance between the fine-tuned and pre-trained weights. Formally,

$$\mathcal{L}(\theta) = \tilde{\mathcal{L}}(\theta) + \frac{\lambda}{2}\|\theta - \theta_0\|_2^2 \tag{1}$$

where $\theta$ denotes the fine-tuned weights, $\theta_0$ the pre-trained weights, $\tilde{\mathcal{L}}(\theta)$ the original loss function, and $\lambda$ the hyper-parameter for regularization strength, i.e., weight decay. In this case, $\|\theta - \theta_0\|_2^2$ serves as a constraint so that the updated model will not deviate from the initialization too much, thus we can maintain some strengths from the pre-trained model. However, L2-SP is not intuitive to tune $\lambda$ which often spans over a wide range: a small $\lambda$ may achieve better in-distribution performance but leads to poor OOD robustness, while a large $\lambda$ results in underfitting. Besides, L2-SP is also harder to tune if applied differently across layers.

In this work, we instead propose to view robust fine-tuning from a *multi-objective optimization* perspective, leading to a more explicit method to balance this trade-off. Specifically, there are two objectives that we want to optimize,

$$\textbf{Objective}_1 = \tilde{\mathcal{L}}(\theta), \textbf{Objective}_2 = \frac{1}{2}\|\theta - \theta_0\|_2^2 \tag{2}$$

where the first objective represents the original loss function and the second objective represents the distance between the fine-tuned and pre-trained weights. Our goal is to minimize both at the same time to achieve ID generalization and OOD robustness.

## 3.2   PROJECTING CONFLICTING GRADIENTS

Viewed from this perspective, we can leverage prior multi-objective methods towards our problem. PCGrad (Yu et al., 2020) hypothesizes that the key optimization issue in multi-objective learning arises from conflicting gradients, where gradients for different objectives point away from each other. Thus, optimizing one of them will lead to the suboptimality of the others. They propose a form of gradient surgery by projecting a task's gradient onto the normal plane of the gradient of any other task that has a conflicting gradient, therefore benefiting all objectives.

Inspired by PCGrad, we propose the following algorithm for robust finetuning: When the gradients between the two objectives are in conflict, i.e. their cosine similarity is negative, we project the gradient for the original loss function to the orthogonal direction of the gradient for the regularization

term. Specifically, the gradients for the two objectives and the projection of the first gradient in the direction of the second gradient are respectively:

$$\tilde{g}_1 = \nabla_\theta \tilde{\mathcal{L}}(\theta), \ \tilde{g}_2 = \theta - \theta_0, \ \tilde{g}_1^{\text{proj}} = \frac{\tilde{g}_1 \tilde{g}_2}{\|\tilde{g}_2\|^2} \tilde{g}_2 \tag{3}$$

We add a hyper-parameter $\omega \in [0, 1]$ to further control the projection strength. $\omega = 0$ is equivalent to an unconstrained gradient update, while $\omega = 1$ is the same as a full orthogonal projection. The final projected gradient is the following:

$$g = \tilde{g}_1 - \omega \tilde{g}_1^{\text{proj}} = \tilde{g}_1 - \omega \frac{\tilde{g}_1 \tilde{g}_2}{\|\tilde{g}_2\|^2} \tilde{g}_2, \ \omega \in [0, 1] \tag{4}$$

Note that for L2-SP, the gradient for the regularized loss function is formulated similarly:

$$g = \nabla_\theta \mathcal{L}(\theta) = \nabla_\theta \tilde{\mathcal{L}}(\theta) + \lambda(\theta - \theta_0) = \tilde{g}_1 + \lambda \tilde{g}_2 \tag{5}$$

Thus, *DiGraP* is equivalent to L2-SP with different $\lambda$ for every layer. In summary, for each layer $i$:

- **Gradients are non-conflicting** ($\tilde{g}_1^i \tilde{g}_2^i \geq 0$): $\lambda^i = 0$
- **Gradients are conflicting** ($\tilde{g}_1^i \tilde{g}_2^i < 0$): $\lambda^i = -\omega^i \frac{\tilde{g}_1^i \tilde{g}_2^i}{\|\tilde{g}_2^i\|^2}$

Compared to L2-SP, the hyper-parameter $\omega$ in *DiGraP* is within the range between 0 and 1, which is more intuitive to tune. Furthermore, even with one fixed $\omega$, the regularization strength $\lambda$ varies across both layers and iterations, making the fine-tuning process more flexible to fit the training data.

### 3.3 LAYER-WISE TRAINABLE DIRECTIONAL GRADIENT PROJECTION

We emphasize that the regularization problem in Eq. 1 is still not fully equivalent to the multi-objective optimization in Eq. 2. Specifically, for a multi-objective optimization problem, we want to optimize all objective functions, i.e., to minimize both $\tilde{\mathcal{L}}(\theta)$ and $\frac{1}{2}\|\theta - \theta_0\|_2^2$ in our case. However, for a regularization problem, the regularization term does not necessarily decrease. Instead, it acts as a constraint on the original loss function and the regularization term is smaller compared to the one in a model trained without regularization. Projecting the original gradient to the orthogonal direction of the gradient for the regularization term will potentially lead to underfitting. It is especially detrimental at the beginning of the training, where the fine-tuned weights are close to the pre-trained weights, thus it is more beneficial for the model to stick to its original gradient descent direction.

As a result, we aim for the projection strength $\omega$ to be dynamic throughout the training process. Intuitively, $\omega$ should start small during the early iterations, allowing the model to prioritize fitting to the downstream task. As training progresses and the fine-tuned model diverges further from its initial state, $\omega$ should gradually increase to guide the fine-tuned gradient direction towards alignment with the regularization gradient direction. In Sec. 5.2 we will visualize the variation of projection strength $\omega$ throughout training to further validate this motivation.

To achieve this, we make the projection strength $\omega$ trainable, allowing it to adapt throughout the training process. For the $t$ step of unconstrained gradient descent with the learning rate of $\alpha$, the model weights update as follows,

$$\tilde{\theta}_t = \theta_{t-1} - \alpha \tilde{g}_{t,1} \tag{6}$$

where $\tilde{\theta}_t$, $\theta_{t-1}$ and $\tilde{g}_{t,1}$ denote the unconstrained model weights at current step $t$, the model updates of previous step $t-1$ and the gradient for the original loss function at current step $t$.

For one step of directional gradient descent, the model weights update as follows,

$$\theta_t = \theta_{t-1} - \alpha(\tilde{g}_{t,1} - \omega_t \frac{\tilde{g}_{t,1} \tilde{g}_{t,2}}{\|\tilde{g}_{t,2}\|^2} \tilde{g}_{t,2}) = \tilde{\theta}_t + \alpha \omega_t \frac{\tilde{g}_{t,1} \tilde{g}_{t,2}}{\|\tilde{g}_{t,2}\|^2} \tilde{g}_{t,2} \tag{7}$$

where $\theta_t$, $\omega_t$ and $\tilde{g}_{t,2}$ denote the constrained model weights, the projection strength and the gradient for the regularization term at current step $t$. $\tilde{\theta}_t$ and $\tilde{g}_{t,1}$ are the same as the ones in Eq. 6.

The derivative of the original loss function $\tilde{\mathcal{L}}(\theta)$ w.r.t. $\omega$ is as follows using the chain rule:

$$\nabla\omega := \frac{\partial\tilde{\mathcal{L}}(\theta_{t-1})}{\partial\omega} = \frac{\partial\tilde{\mathcal{L}}(\theta_{t-1})}{\theta}\frac{\partial\theta_{t-1}}{\partial\omega} = \alpha\tilde{g}_{t,1}\tilde{g}_{t-1,1}^{\text{proj}} \qquad (8)$$

We initialize $\omega_0 = 0$ for the first iteration and update with learning rate of $\mu$. We also add normalization on $\nabla\omega$ for numerical stability. For *DiGraP*, instead of tuning the weight decay $\lambda$ in L2-SP, we only tune the learning rate $\mu$ of the projection strength $\omega$. We argue that tuning $\mu$ is less sensitive and will provide sensitivity analysis in Sec. 5.2. We also compare fixed and trainable projection strength $\omega$ in Sec. 5.3. The final algorithm of *DiGraP* is illustrated in Alg. 1.

---

**Algorithm 1** Adam with Trainable Directional Gradient Projection

---

**Input:** $\theta_0$: pre-trained model, $\alpha$: learning rate, $\mu$: learning rate for $\omega$, $(\beta_1, \beta_2) \leftarrow (0.9, 0.999)$
**Initialize:** $m_0 \leftarrow 0$, $v_0 \leftarrow 0$
**for** $t = 1$ to $T$ **do**
$\quad\begin{cases}\tilde{g}_{t,1} \leftarrow \nabla_\theta\mathcal{L}(\theta_{t-1}) \\ \tilde{g}_{t,2} \leftarrow \theta_{t-1} - \theta_0\end{cases}$ $\qquad\qquad\qquad\qquad$ ▷ Gradients of the Objectives (Eq. 3)
$\quad\tilde{g}_{t,1}^{\text{proj}} \leftarrow \frac{\tilde{g}_{t,1}\tilde{g}_{t,2}}{\|\tilde{g}_{t,2}\|^2}\tilde{g}_{t,2}$ $\qquad\qquad\qquad\qquad$ ▷ Gradient Projection (Eq. 3)
$\quad$**if** $t = 1$ **then**
$\quad\quad\omega_t \leftarrow 0$ $\qquad\qquad\qquad\qquad\qquad\qquad\qquad\qquad$ ▷ Initialize $\omega$
$\quad$**else**
$\quad\quad\begin{cases}\nabla\omega_t \leftarrow \textbf{Normalization}(\alpha_{t-1}\tilde{g}_{t,1}\tilde{g}_{t-1,1}^{\text{proj}}) \\ \omega_t \leftarrow \max(0, \min(1, \textbf{AdamUpdate}(\omega_{t-1}, \nabla\omega_t, \mu, t))\end{cases}$ $\qquad$ ▷ Updating $\omega$ (Eq. 8)
$\quad$**if** $\tilde{g}_{t,1}\tilde{g}_{t,2} < 0$ **then**
$\quad\quad g_t \leftarrow \tilde{g}_{t,1}$ $\qquad\qquad\qquad\qquad$ ▷ Unconstrained Gradient Descent (Eq. 6)
$\quad$**else**
$\quad\quad g_t \leftarrow \tilde{g}_{t,1} - \omega_t\tilde{g}_{t,1}^{\text{proj}}$ $\qquad\qquad$ ▷ Directional Gradient Projection (Eq. 4, Eq. 7)
$\quad m_t \leftarrow \beta_1 m_{t-1} + (1 - \beta_1)g_t$
$\quad v_t \leftarrow \beta_2 v_{t-1} + (1 - \beta_2)g_t^2$
$\quad$**Bias Correction:** $\widehat{m}_t \leftarrow \frac{m_t}{1-\beta_1^t}$, $\widehat{v}_t \leftarrow \frac{v_t}{1-\beta_2^t}$
$\quad$**Update:** $\theta_t \leftarrow \theta_{t-1} - \frac{\alpha_t\widehat{m}_t}{\sqrt{\widehat{v}_t}+\epsilon}$

---

In summary, our unique contribution lies in adapting gradient projection to the specific challenges of robust fine-tuning. While PCGrad was designed for multi-task learning, *DiGraP* extends these principles to the fine-tuning of pre-trained models by introducing a hyper-optimizer and tailoring gradient projection to balance both ID and OOD performance. This refinement allows us to address the unique trade-offs in robust fine-tuning, which are distinct from those in multi-task optimization.

### 3.4 COMPATABILITY WITH PARAMETER-EFFICIENT FINE-TUNING (PEFT) METHODS

*DiGraP* is further compatible with PEFT methods such as LoRA (Hu et al., 2021), which is a prevalent fine-tuning strategy for large foundation models. PEFT methods generally update new parameters to add to the original weights. In this case, instead of optimizing the distance between fine-tuned and pre-trained weights $\frac{1}{2}\|\theta - \theta_0\|^2$, we minimize the distance between the updated weight and origin $\frac{1}{2}\|\theta\|^2$ in PEFT. Thus, when combined with PEFT, *DiGraP* does not need to save an additional pre-trained copy and requires the same amount of memory as PEFT. In Sec. 4.2 and Sec. 4.3, we demonstrate that *DiGraP* can further improve the results of LoRA on VQA tasks.

## 4 EXPERIMENTS

**Overview.** We test *DiGraP* on a variety of benchmarks, tasks and architectures to validate its effectiveness. The experiments are split into three sections including image classification (Sec. 4.1), reformulating image classification as VQA tasks (Sec. 4.2) and fine-tuning on VQA datasets (Sec. 4.3). We emphasize that it is important to move robust fine-tuning towards multi-modal models, given their

Table 1: **DomainNet Results using MOCO-V3 pre-trained ResNet50 with Real Data.** *DiGraP* outperforms baselines on average OOD. **Bold**: best. Underline: second best.

| | ID | OOD | | | | | Statistics | |
|---|---|---|---|---|---|---|---|---|
| | Real | Sketch | Painting | Infograph | Clipart | OOD Avg. | ID Δ (%) | OOD Δ (%) |
| Vanilla FT | 81.99 | 31.52 | 42.89 | 18.51 | 44.98 | 34.47 | 0.00 | 0.00 |
| Linear Prob. | 73.01 | 24.10 | 39.56 | 12.27 | 30.38 | 26.58 | -10.96 | -22.90 |
| Partial Fusion | 78.27 | 27.72 | 39.74 | 15.56 | 38.18 | 30.30 | -4.55 | -12.11 |
| L2-SP | 81.51 | 34.91 | 45.76 | 18.97 | 45.29 | 36.23 | -0.59 | 5.09 |
| MARS-SP | 81.89 | 34.44 | 45.05 | 19.97 | 46.36 | 36.45 | -0.13 | 5.74 |
| LP-FT | **82.92** | 34.50 | 45.42 | 20.12 | 47.11 | 36.79 | **1.13** | 6.72 |
| TPGM | 82.66 | 35.35 | 46.20 | 20.13 | 45.75 | 36.86 | 0.82 | 6.91 |
| FTP | 82.17 | 36.26 | 46.58 | 20.67 | 46.97 | 37.62 | 0.22 | 9.13 |
| DiGraP (0.1) | 82.20 | **36.43** | **46.75** | **21.40** | **47.46** | **38.01** | 0.26 | **10.27** |

popularity and richness in terms of different types and strenghts of distribution shift. For Sec. 4.1, we use a discriminative backbone - ImageNet pretrained MOCO-V3 ResNet50 (Chen et al., 2021) as the pre-trained model. We follow the setting of the previous work and further details can be found at Tian et al. (2023a). For Sec. 4.2 and Sec. 4.3, we use a generative backbone - Google's recently released PaliGemma (Beyer et al., 2024) pretrained on a broad mixture of large-scale vision-language tasks, which is lightweight and achieves state-of-the-art performance on VQAv2. In Appendix 7.1, we include additional results with more backbones and a langugae-only experiment.

**Datasets.** For Sec. 4.1 and Sec. 4.2, we use DomainNet (Peng et al., 2019) as the benchmark, which consists of six domains (real, sketch, painting, infograph, clipart and quickdraw) with 345 classes. We fine-tune our model on real domain and evaluate on all other domains. For Sec. 4.3, we fine-tune on VQAv2 (Goyal et al., 2017) and test on nine OOD datasets using LoRA (Hu et al., 2021). For the near OODs, we evaluate on VQAv2's six variants, namely IV-VQA (Agarwal et al., 2020), CV-VQA (Agarwal et al., 2020), VQA-Rephrasings (Shah et al., 2019), VQA-CP v2 (Agrawal et al., 2018), VQA-CE (Dancette et al., 2021) and AdVQA (Sheng et al., 2021), which cover uni-modal, multi-modal and adversarial distribution shifts from VQAv2. We also include TextVQA (Singh et al., 2019), VizWiz (Bigham et al.) and OK-VQAv2 (Reichman et al., 2023), which are constructed from different sources than VQAv2, as the far OOD datasets. Further details can be found in Sec. 4.3.

**Training Details.** The hyper-parameter $\mu$ is found through cross-validation per dataset, and the model with the best ID validation accuracy is taken. We leave all training details to Appendix 7.2

### 4.1 IMAGE CLASSIFICATION EXPERIMENTS

***DiGraP* outperforms robust fine-tuning baselines on image classification task.** We utilize the DomainNet as benchmark and compare *DiGraP* with several existing methods using ImageNet pre-trained MOCO-V3 ResNet50 as initialization. We follow the training scheme and use the same hyper-parameters of prior work (Tian et al., 2023a). In Tab. 1, we observe that *DiGraP* achieves the best OOD performance across all OOD domains and a competitive ID performance on the real domain. Specifically, *DiGraP* outperforms L2-SP (Li et al., 2018) and magnitude-based projection methods (Tian et al., 2023a;b) on both ID and OOD results. Note that we use the reported baseline results from Tian et al. (2023b) where the Quickdraw results are not presented.

### 4.2 REFORMULATING IMAGE CLASSIFICATION AS VQA TASKS

Previous work primarily focuses on uni-modal benchmarks but is seldom tested on multi-modal settings. As an additional contribution, we first bridge the gap by using the same benchmark but reformulate it as a VQA task and tested several robust fine-tuning methods. Specifically, inspired by Ging et al. (2024), we change DomainNet to DomainNet-oVQA by using the same images but asking questions such as "What is in the image?" with class labels as the ground truth answers. To make the two tasks more comparable, we use the ClipMatch (ClipM) metric from Ging et al. (2024) by embedding the model prediction and class names with EVA-Clip (Sun et al., 2023) and obtain the most similar one by matching them using cosine similarity. We consider this benchmark as one OOD dataset with distribution shifts only in image modality and will conduct a more comprehensive

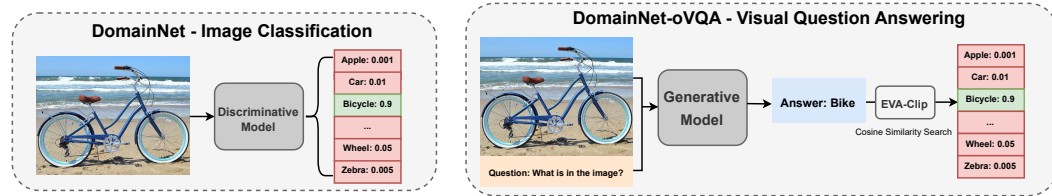

Figure 2: **Reformulating Image Classification as VQA.** (Left) DomainNet as conventional Image Classification tasks. (Right) DomainNet-oVQA as VQA tasks. We use the same images but add questions "What is in the image?" to generate answers and use original labels as the ground truth answers. To make the two tasks more comparable, we feed the generated answers and 345 class labels into the EVA-Clip text encoder and use cosine similarity search to obtain the most similar class.

experiments with distribution shifts in other modalities in Sec. 4.3. We use the pre-trained generative vision-language model (VLM) PaliGemma (Beyer et al., 2024) as initialization.

Table 2: **DomainNet-oVQA Fine-Tuning Results.** PaliGemma-3B fine-tuned on DomainNet-oVQA (Real) and evaluated on other domains as a VQA task. We use LoRA for efficiency. Note that L2-SP reduces to Vanilla FT with AdamW under LoRA. *DiGraP* achieves SOTA results on both ID and OOD performance. **Bold**: best. Underline: second best.

| | ID | OOD | | | | | Statistics | | |
| | Real | Sketch | Painting | Infograph | Clipart | Quickdraw | OOD Avg. | ID Δ (%) | OOD Δ (%) |
|---|---|---|---|---|---|---|---|---|---|
| Zero-Shot | 64.09 | 39.00 | 39.80 | 28.87 | 57.86 | 3.27 | 33.76 | - | - |
| Vanilla FT[2] | 92.57 | 70.65 | 71.17 | 54.75 | 82.68 | 18.73 | 59.60 | 0.00 | 0.00 |
| Linear Prob. | 91.10 | 68.52 | 67.23 | 49.86 | 79.78 | 19.74 | 57.03 | -1.59 | -4.31 |
| LP-FT | 92.60 | 71.13 | 70.64 | 54.66 | 81.41 | **20.94** | 59.76 | 0.03 | 0.27 |
| WiSE-FT | 80.57 | 64.70 | 65.08 | 43.89 | 72.98 | 17.80 | 52.89 | -12.96 | -11.26 |
| FTP | 87.49 | 71.38 | 69.33 | 53.33 | 79.18 | 19.74 | 58.59 | -5.49 | -1.69 |
| DiGraP (0.5) | **92.72** | **72.56** | **72.31** | **57.52** | **83.32** | 20.13 | **61.17** | **0.16** | **2.63** |

**Vanilla fine-tuning outperforms zero-shot on both ID and OOD datasets.** Surprisingly, vanilla fine-tuning on DomainNet-Real-oVQA improves the performance on every domain in DomainNet-oVQA, while doing the same but as an image classification task degrades the OOD performance compared to zero-shot (Wortsman et al., 2022). Note that during fine-tuning in the image classification task, we only take the image encoder from the backbone model and add a linear head after it, i.e., we remove the text encoder and uses cross-entropy loss which is different from the pre-trained loss used during pre-training. However, (Goyal et al., 2023) points out that fine-tuning is more robust when trained with the same objective as pre-training. This may be a potential reason for the robustness of vanilla fine-tuning for VQA tasks, since fine-tuning on the VQAs does not change the pre-training model structure and loss function. We will not focus on this problem in this paper and leave further discussion to future work. Nevertheless, one interesting question remains: *when vanilla fine-tuning performs well on datasets with distributon shifts, can previous robust fine-tuning baselines further increase the robustness?*

**WiSE worsens both ID generalization and OOD robustness under VQA tasks.** Wortsman et al. (2022) reports that ensembling the weights of the zero-shot and fine-tuned models can benefit from both the robustness of the zero-shot and the downstream adaptation of the fine-tuned models for image classification tasks. However, WiSE is highly dependent on the performance of the pre-trained model, and only works when vianilla fine-tuning decreases the robustness. In Tab. 2, there is a huge gap between the zero-shot and fine-tuned results. In this case, naively combining the pre-trained weights reduces the model's robustness under all distribution shifts compared to vanilla fine-tuning.

**Directional Gradient Projection benefits the models in general.** In Tab. 2, *DiGraP* outperforms two-stage training (LP-FT), weight interpolation (WiSE-FT) and bi-level optimization (FTP) on both ID and average OOD performance. We argue that WiSE-FT only works for the setting where zero-shot performance is strong while *DiGraP* is beneficial for general cases.

---

[2]Same as L2-SP (Li et al., 2018) under LoRA (Hu et al., 2021)

## 4.3 Fine-tuning on VQA Datasets

We now conduct a comprehensive experiment on various VQA datasets with different distribution shifts. We consider VQAv2 (Goyal et al., 2017) as the ID dataset. We further evaluate the model on six near OOD datasets which construct different types of distribution shifts from VQAv2 and three far OOD datasets in which the image and text sources are completely different from VQAv2.

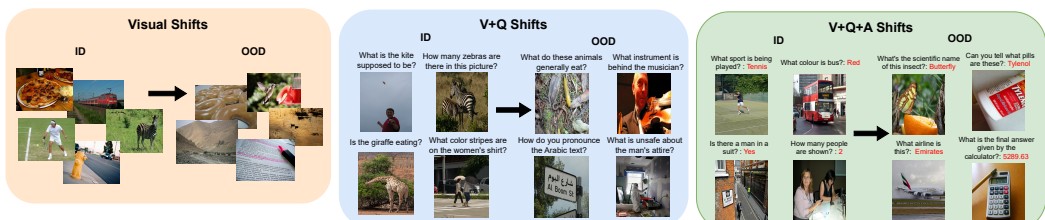

Figure 3: **ID and OOD VQA datasets with Uni-modal and Multi-modal Distribution Shifts.**

**ID Dataset.** VQAv2 (Goyal et al., 2017) builds upon VQAv1 (Agrawal et al., 2017) by balancing question-answer pairs with complementary images to minimize the bias in language priors, thus is more challenging and is widely used as a benchmark for popular vision-language models.

**OOD Datasets.** 1) *Distribution Shifts to Images.* IV-VQA (Agarwal et al., 2020) and CV-VQA (Agarwal et al., 2020) remove the objects irrelevant to answering the question and generate complementary images with one instance of the object removed respectively. 2) *Distribution Shifts to Questions.* VQA-Rephrasings (Shah et al., 2019) collects three rephrasings of each question. 3) *Distribution Shifts to Answers.* VQA-CP (Agrawal et al., 2018) reorganizes the correlation between the question type and correct answer. 4) *Distribution Shifts to Multi-modalities.* VQA-CE (Dancette et al., 2021) selects a subset of VQAv2 that are counterexamples of potential multi-modal shortcuts. 5) *Adversarial Distribution Shifts.* AdVQA (Sheng et al., 2021) provides human-adversarial examples for questions where the model's predicted answer is incorrect. 6) *Far OODs.* TextVQA (Singh et al., 2019) requires models to answer questions by understanding text embedded in images. VizWiz (Bigham et al.) contains user-generated images with diverse challenges like poor quality, ambiguity, and irrelevant content for answering visual questions. OK-VQAv2 (Reichman et al., 2023) represents a knowledge-based VQA task where the visual question cannot be answered without external knowledge.

**Evaluation and Metrics.** We follow the metric from VQAv2 (Goyal et al., 2017) and the evaluation is based on the accuracy of predicted answers compared to ground truth human-annotated answers. For each question, the dataset includes 10 human-provided answers. The accuracy is calculated as:
$$\text{Accuracy} = \min\left(\frac{\text{number of humans who gave the answer}}{3}, 1\right).$$

**Measuring the OOD Distance.** We explore shifts on single image modality and joint image question (V+Q) shifts, as well as image question answer (V+Q+A) shifts by computing the test set shift relative to the training domain (i.e. VQAv2 train) using the negative Mahalanobis distance metric. The higher the value, the less the distribution shift. More details are in Appendix 7.3.

**Experimental Results.** We fine-tune the PaliGemma-3B model on the VQAv2 dataset with LoRA and evaluate on the other OOD datasets. The results of *DiGraP* and other robust fine-tuning methods are shown in Tab. 3. We have the following observations.

**Smaller distribution shifts correlate with better OOD performance.** The analysis of image and joint shifts reveals a high correlation with VQA performance, evidenced by correlation values of 0.83 and 0.80, respectively. This suggests that larger shifts significantly degrade VQA performance. Such trends validate our methodology in quantifying shifts, as far OOD scenarios align with increased shift levels and diminished performance.

**Full fine-tuning improves zero-shot performance across ID, near OOD, and far OOD datasets.** As shown in Tab. 3, we observe no degradation in OOD performance following vanilla fine-tuning, even when PaliGemma is pre-trained on VQA tasks. This may be attributed to reasons similar to

---

[3]Same as L2-SP (Li et al., 2018) under LoRA (Hu et al., 2021)

Table 3: **Visual Question Answering Fine-Tuning Results using PaliGemma-3B.** *DiGraP* outperforms baselines across ID and near OOD and is competitive on far OOD datasets using LoRA. Note that Vanilla FT with AdamW reduces to L2-SP under LoRA. Trainable projection strength **Bold**: best. Underline: second best.

| | ID | Near OOD | | | | | | | Far OOD | | | | |
| | VQAv2 (val) | Vision | | Question VQA-Rep. | Answer VQA-CP v2 | Multimodal VQA-CE | Adversarial AdVQA | Avg. | TextVQA | VizWiz | OK-VQA | Avg. | OOD Avg. |
| | | IV-VQA | CV-VQA | | | | | | | | | | |
|---|---|---|---|---|---|---|---|---|---|---|---|---|---|
| Zero-Shot | 54.42 | 63.95 | 44.72 | 50.10 | 54.29 | 30.68 | 30.46 | 45.70 | 14.86 | 16.84 | 28.60 | 20.10 | 37.17 |
| Vanilla FT[3] | 86.29 | 94.43 | **69.36** | 78.90 | 86.21 | 71.73 | 49.82 | 75.08 | 42.08 | 22.92 | 48.30 | 37.77 | 62.64 |
| Linear Prob. | 78.24 | 87.83 | 63.87 | 69.61 | 78.48 | 61.66 | 42.90 | 67.39 | 29.61 | 18.80 | 42.27 | 30.23 | 55.00 |
| LP-FT | 85.97 | 93.30 | 65.93 | 76.49 | 86.16 | 72.73 | 45.68 | 73.38 | 31.41 | 19.01 | 43.27 | 31.23 | 59.33 |
| WiSE-FT | 71.36 | 85.06 | 64.55 | 66.42 | 70.89 | 48.74 | 43.95 | 63.27 | 36.98 | 22.41 | 42.35 | 33.91 | 53.48 |
| FTP | 81.77 | 92.61 | 67.93 | 76.66 | 81.41 | 64.14 | 50.99 | 72.29 | **49.12** | **25.67** | **51.07** | **41.95** | 62.18 |
| DiGraP (0.5) | **87.40** | **95.16** | 68.56 | **79.29** | **87.19** | **73.63** | **51.41** | **75.87** | 44.12 | 22.98 | 49.20 | 38.77 | **63.50** |
| Vision Shift | 27.84 | 27.17 | 28.67 | 27.94 | 27.92 | 27.91 | 27.66 | 27.88 | 28.93 | 32.92 | 27.98 | 29.94 | 28.57 |
| Question Shift | 38.70 | 37.08 | 28.70 | 40.84 | 38.84 | 40.19 | 40.68 | 37.72 | 47.10 | 46.25 | 48.20 | 47.18 | 40.85 |
| Joint Shift | 34.81 | 32.91 | 29.55 | 35.26 | 34.81 | 37.10 | 34.97 | 34.10 | 45.72 | 45.69 | 40.91 | 44.11 | 37.44 |

those discussed in Sec. 4.2, or due to differing characteristics of the backbone models and tasks. Once again, WiSE negatively impacts both ID generalization and OOD robustness in VQA tasks when interpolating between pre-trained and fine-tuned weights.

***DiGraP* outperforms baselines on both ID and near-OOD datasets.** Beyond improvements on uni-modal tasks, *DiGraP* enhances vanilla fine-tuning and consistently outperforms other baselines in multi-modal settings. As shown in Tab. 3, *DiGraP* achieves the highest ID and average near-OOD results, demonstrating robustness to distribution shifts across various modalities, including vision, question, answer, their combinations, and adversarial shifts.

***DiGraP* is competitive on far OOD datasets.** From Tab. 3, we observe that *DiGraP* also improves vanilla fine-tuning on the three far OOD benchmarks and is the second best among all robust fine-tuning methods. Notably, while FTP (Tian et al., 2023a) performs well on OOD, it severely underfits the training domain with a substantial lower ID performance compared to vanilla fine-tuning and *DiGraP*, even with the positive gradient annealing factor $\kappa = 0$, which indicates weakest regularization. However, FTP demonstrates outstanding performance on far OOD tasks with significant higher results on the three far OOD datasets. One potential reason is that FTP imposes much stronger regularization since even when $\kappa = 0$ the projection constraints are *non-decreasing* during training, which means it still provides regularization. In FTP, the authors also mention that $\kappa = 0$ is necessary to obtain the best performance if underfitting is observed. However, in *DiGraP*, $\omega = 0$ is equivalent to unconstrained fine-tuning with no regularization. Thus, *DiGraP* enforces weaker regularization than FTP. We emphasize that the significant performance decrease from zero-shot to fine-tuning is mostly observed on rather simple tasks (e.g. image classification), while *DiGraP* is more general for all cases. Nevertheless, it remains interesting why FTP significantly increases the far OOD performance given the zero-shot is poor in Tab. 3, and we will leave the exploration to future work.

# 5 HYPER-PARAMETER TUNING AND ABLATION STUDY

## 5.1 VARIATION OF PROJECTION STRENGTH THROUGHOUT TRAINING

We visualize the variation of the average projection strength $\omega$ of all layers over iterations for five different hyper-parameters $\mu \in \{0.01, 0.1, 0.5, 1, 100\}$ in Fig. 4. As we increase $\mu$, the projection strength $\omega$ becomes larger. For all cases, the projection strength $\omega$ starts from zero and converges at the end of the training. This aligns with our intuition that the projection strength should vary over time to learn dynamic priority of the two objectives during different stage of training.

## 5.2 IMPACT OF HYPER-PARAMETER SENSITIVITY ON ROBUSTNESS

We further perform the sensitivity analysis of the hyper-parameter $\mu$ on ID and average OOD performance for DomainNet-oVQA and VQA experiments. Results from Tab. 4 show that both ID and OOD results fluctuate slightly even when $\mu$ spans over a wide range from 0.01 to 100. This again proves that *DiGrap* is more controllable and less sensitive to hyper-parameter change.

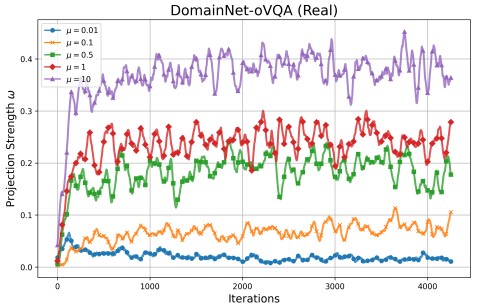 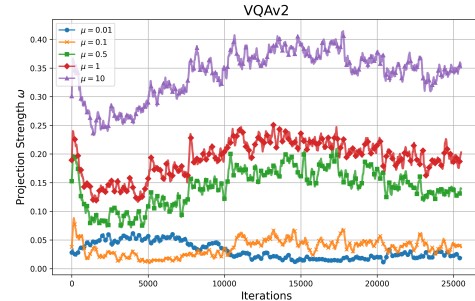

Figure 4: Variation of the Average Projection Strength $\omega$ of all Layers over Iterations. We present the results of $\mu \in \{0.01, 0.1, 0.5, 1, 100\}$. We use the sliding window with window sizes of 50 and 200 to visualize the results for DomainNet-oVQA (Real) and VQAv2. The projection strength $\omega$ is dynamic over iterations, growing from small to large and converging in the end.

Table 4: **Sensitivity Analysis of Hyper-Parameter $\mu$ on ID and OOD performance.** We sweep $\mu \in \{0.01, 0.1, 0.5, 1, 10\}$. For DomainNet-oVQA and VQA experiments, both ID and average OOD performance fluctuates slightly and are robust to the change of $\mu$ over a wide range.

| Hyper-Parameter $\mu$ | 0.01 | 0.1 | 0.5 | 1 | 100 |
|---|---|---|---|---|---|
| OOD Avg. | 60.95 | 59.96 | 61.17 | 60.31 | 60.40 |
| ID | 92.52 | 92.68 | 92.72 | 92.65 | 92.45 |

(a) DomainNet-oVQA hyper-parameter ($\mu$) sweep.

| Hyper-Parameter $\mu$ | 0.01 | 0.1 | 0.5 | 1 | 100 |
|---|---|---|---|---|---|
| OOD Avg. | 63.15 | 62.78 | 63.50 | 63.52 | 63.44 |
| ID | 86.91 | 86.85 | 87.40 | 87.08 | 87.18 |

(b) VQA hyper-parameter ($\mu$) sweep.

Table 5: **Comparing Fixed and Trainable Projection Strength $\omega$ on DomainNet-oVQA. Bold**: best. Underline: second best. Trainable projection strength outperforms different fixed projection strengths on both ID and average OOD.

| | ID | OOD | | | | | OOD Avg. |
|---|---|---|---|---|---|---|---|
| | Real | Sketch | Painting | Infograph | Clipart | Quickdraw | |
| DiGraP ($\omega = 0.1$) | 92.49 | 71.86 | 71.36 | 56.34 | 83.15 | 19.22 | 60.39 |
| DiGraP ($\omega = 0.5$) | 92.63 | 71.24 | 71.46 | 56.08 | 83.17 | 18.61 | 60.11 |
| DiGraP ($\omega = 0.9$) | 92.53 | **73.09** | 71.73 | 56.34 | 82.85 | 20.11 | 60.82 |
| DiGraP (trainable) | **92.72** | 72.56 | **72.31** | **57.52** | **83.32** | **20.13** | **61.17** |

### 5.3 ABLATING FIXED AND TRAINABLE PROJECTION STRENGTH

To validate the effectiveness of trainable projection strength $\omega$, we conduct analysis to compare with fixed projection strength with different values $\omega \in \{0.1, 0.5, 0.9\}$. Tab. 5 shows that trainable *DiGraP* outperforms the others and achieves the best ID and average OOD results.

## 6 CONCLUSION

We present Directional Gradient Projection (*DiGraP*), a novel method for robust fine-tuning that leverages gradient-based directional information to unify regularization and multi-objective optimization. *DiGraP* addresses hyperparameter sensitivity and underfitting issues in existing methods. Experiments on image classification and VQA benchmarks show that *DiGraP* surpasses baselines, improving ID accuracy and OOD robustness, while bridging uni-modal and multi-modal evaluation for robust fine-tuning across domains. However, *DiGraP* struggles with far OOD datasets due to limited regularization, excelling in near OOD scenarios but facing a trade-off as stronger regularization may harm ID and near OOD performance (Sec. 4.2, 4.3). Future work should balance ID, near OOD, and far OOD performance. Additionally, *DiGraP* is suited for fine-tuning well pre-trained models, with its efficacy in training from scratch or non-robust initialization yet to be explored.

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

## 7 Appendix

### 7.1 Additional Results

**1) More Backbones: CLIP, LLaVA.** We include experiments of fine-tuning on DomainNet with CLIP ViT-Base (Radford et al., 2021) (Tab. 8), DomainNet-oVQA (Tab. 7), and VQA (Tab. 6) with LLaVA-7B (Liu et al., 2023). *DiGraP* consistently achieves the best performance across all experiments.

Table 6: **Visual Question Answering Fine-Tuning Results using LLaVA-7B (Liu et al., 2023).** We sample 10% of the VQAv2 training and validation set. We fine-tune using LoRA with a rank of 4 and target on the $W_q, W_v$. *DiGraP* outperforms baselines across ID and near OOD and is competitive on far OOD datasets using LoRA. Note that Vanilla FT with AdamW reduces to L2-SP under LoRA. Trainable projection strength **Bold**: best. Underline: second best.

| | ID | Near OOD | | | | | | | Far OOD | | | | |
|---|---|---|---|---|---|---|---|---|---|---|---|---|---|
| | VQAv2 (val) | Vision | | Question VQA-Rep. | Answer VQA-CP v2 | Multimodal VQA-CE | Adversarial AdVQA | Avg. | TextVQA | VizWiz | OK-VQA | Avg. | OOD Avg. |
| | | IV-VQA | CV-VQA | | | | | | | | | | |
| Zero-Shot | 3.27 | 6.34 | 4.40 | 2.92 | 4.28 | 1.46 | 1.22 | 3.44 | 1.10 | 0.24 | 0.71 | 0.68 | 2.52 |
| Vanilla FT[4] | 72.49 | **82.23** | **58.61** | 63.93 | 69.80 | 45.60 | 40.22 | 60.07 | 37.16 | 12.11 | 36.74 | 28.67 | 49.60 |
| LP-FT | 53.01 | 39.26 | 38.54 | 27.93 | 33.14 | 9.24 | 23.66 | 28.63 | 7.80 | 5.16 | 9.95 | 7.64 | 21.63 |
| WiSE-FT | 60.47 | 63.98 | 46.39 | 50.26 | 55.79 | 20.23 | 23.35 | 43.33 | 10.10 | 3.15 | 13.97 | 9.07 | 31.98 |
| FTP | 67.95 | 80.65 | 57.33 | 61.66 | 68.05 | 43.70 | 39.53 | 58.49 | 33.88 | **12.98** | 31.77 | 26.21 | 47.73 |
| DiGraP (0.01) | **72.54** | 83.64 | 56.56 | **64.70** | **69.71** | **45.15** | **43.10** | **60.48** | **38.22** | 12.97 | **37.43** | **29.54** | **50.17** |

Table 7: **DomainNet-oVQA Fine-Tuning Results.** LLaVA-7B (Liu et al., 2023) fine-tuned on DomainNet-oVQA (Real) and evaluated on other domains as a VQA task. We use LoRA for efficiency . Note that L2-SP reduces to Vanilla FT with AdamW under LoRA. *DiGraP* achieves SOTA results on both ID and OOD performance. **Bold**: best. Underline: second best.

| | ID | OOD | | | | | Statistics | | |
|---|---|---|---|---|---|---|---|---|---|
| | Real | Sketch | Painting | Infograph | Clipart | Quickdraw | OOD Avg. | ID Δ (%) | OOD Δ (%) |
| Zero-Shot | 67.71 | 53.45 | 56.15 | 37.90 | 58.32 | 15.87 | 44.34 | - | - |
| Vanilla FT[5] | **86.03** | 63.93 | 59.47 | 45.50 | 72.85 | 15.94 | 51.54 | 0.00 | 0.00 |
| Linear Prob. | 76.84 | 51.41 | 50.53 | 32.67 | 60.86 | 8.72 | 40.84 | -10.68 | -20.76 |
| LP-FT | 79.76 | 56.12 | 55.77 | 37.10 | 66.03 | 12.82 | 45.57 | -7.29 | -11.58 |
| FTP | 77.70 | 55.96 | 43.74 | 39.05 | 62.61 | 15.78 | 43.43 | -9.68 | -15.74 |
| DiGraP (0.1) | 85.68 | **64.40** | **60.26** | **45.94** | **72.95** | **16.69** | **52.05** | -0.41 | 0.98 |

Table 8: **OOD Results on DomainNet**. CLIP ViT-Base finetuned on DomainNet (Real) dataset and evaluated on other domains. **Bold**: best. Underline: second best.

| Method | ID | OOD | | | | Statistics | | |
|---|---|---|---|---|---|---|---|---|
| | Real | Sketch | Painting | Infograph | Clipart | OOD Avg. | ID Δ (%) | OOD Δ (%) |
| Vanilla FT | 86.5 | 45.9 | **58.9** | 34.8 | 59.7 | 49.82 | 0 | 0 |
| LP | 83.2 | 38.6 | 51.4 | 27.2 | 50.7 | 41.97 | -3.82 | -15.76 |
| L2-SP | 86.4 | 46.0 | **58.9** | **35.0** | 59.8 | 49.92 | -0.12 | 0.20 |
| LP-FT | 84.4 | 36.5 | 50.5 | 27.2 | 51.4 | 41.40 | -2.43 | -16.90 |
| FTP | **86.7** | 49.1 | 55.8 | 32.2 | 61.4 | 49.63 | 0.23 | -0.38 |
| DiGraP (1) | 86.2 | **51.3** | 57.3 | 34.0 | **62.4** | **51.25** | -0.35 | 2.87 |

**2) Language-only Experiments.** We also conduct a language-only experiment in Tab. 9, where we use BOSS (Yuan et al., 2023), an NLP benchmark suite designed for OOD robustness evaluation. This suite includes both ID and OOD language-only datasets across multiple tasks (e.g., Sentiment Analysis, Toxic Detection).

**3) LP-FT Controlled Version.** In Tab. 10, we conduct an LP-FT-C experiment (a controlled version of LP-FT), similar to TPGM-C in Tian et al. (2023a), by increasing the regularization strength to ensure that the ID performance matches that of DiGraP. Despite this adjustment, *DiGraP* still outperforms LP-FT-C on OOD datasets.

---

[4]Same as L2-SP (Li et al., 2018) under LoRA (Hu et al., 2021)
[5]Same as L2-SP (Li et al., 2018) under LoRA (Hu et al., 2021)

Table 9: **BOSS (Yuan et al., 2023) Performance Using T5 (Raffel et al., 2023) Fine-Tuned with AZ.** We leverage BOSS, an NLP benchmark suite for OOD robustness evaluation. We focus on the Sentiment Analysis task which contains Amazon (AZ) as the ID and DynaSent (DS), SemEval and SST as the OOD. **Bold**: best. Underline: second best.

| Dataset | ID AZ | OOD DS | OOD SE | OOD SST | Statistics OOD Avg. | ID $\Delta$ (%) | OOD $\Delta$ (%) |
|---|---|---|---|---|---|---|---|
| Vanilla FT | 85.57 | 43.63 | 48.47 | 67.29 | 53.13 | 0.00 | 0.00 |
| L2-SP | 84.87 | 43.40 | 48.65 | 65.07 | 52.37 | -0.82 | -1.43 |
| FTP | 84.73 | 43.84 | 48.69 | 66.73 | 53.09 | -0.98 | -0.08 |
| DiGraP (8e-3) | **85.60** | **44.10** | **49.07** | **67.39** | **53.52** | **0.04** | **0.73** |

Table 10: **DomainNet Results using MOCO-V3 pre-trained ResNet50 with Real Data.** *DiGraP* outperforms baselines on average OOD. **Bold**: best. Underline: second best.

| | ID Real | OOD Sketch | OOD Painting | OOD Infograph | OOD Clipart | OOD Avg. | Statistics ID $\Delta$ (%) | OOD $\Delta$ (%) |
|---|---|---|---|---|---|---|---|---|
| Vanilla FT | 81.99 | 31.52 | 42.89 | 18.51 | 44.98 | 34.47 | 0.00 | 0.00 |
| Linear Prob. | 73.01 | 24.10 | 39.56 | 12.27 | 30.38 | 26.58 | -10.96 | -22.90 |
| Partial Fusion | 78.27 | 27.72 | 39.74 | 15.56 | 38.18 | 30.30 | -4.55 | -12.11 |
| L2-SP | 81.51 | 34.91 | 45.76 | 18.97 | 45.29 | 36.23 | -0.59 | 5.09 |
| MARS-SP | 81.89 | 34.44 | 45.05 | 19.97 | 46.36 | 36.45 | -0.13 | 5.74 |
| LP-FT | **82.92** | 34.50 | 45.42 | 20.12 | 47.11 | 36.79 | **1.13** | 6.72 |
| TPGM | 82.66 | 35.35 | 46.20 | 20.13 | 45.75 | 36.86 | 0.82 | 6.91 |
| FTP | 82.17 | 36.26 | 46.58 | 20.67 | 46.97 | 37.62 | 0.22 | 9.13 |
| LP-FT-C (reg=5) | 82.18 | 35.13 | 46.02 | 20.36 | **47.61** | 37.28 | 0.23 | 8.15 |
| DiGraP (0.1) | 82.20 | **36.43** | **46.75** | **21.40** | 47.46 | **38.01** | 0.26 | **10.27** |

**4) Sensitivity Analysis on Single-Modal Tasks.** In Tab. 11, we include the hyper-parameter tuning experiment for DomainNet on CLIP ViT-Base. The results demonstrate that *DiGraP* remains robust to hyperparameter variations on both ID and OOD datasets in the uni-modal image classification task.

Table 11: **Sensitivity Analysis of Hyper-Parameter $\mu$ on ID and OOD performance.** We sweep $\mu \in \{0.01, 0.1, 0.5, 1, 10\}$. For DomainNet with CLIP ViT-Base experiments, both ID and average OOD performance fluctuates slightly and are robust to the change of $\mu$ over a wide range.

| Hyper-Parameter $\mu$ | 0.01 | 0.1 | 0.5 | 1 | 10 |
|---|---|---|---|---|---|
| OOD Avg. | 50.86 | 51.04 | 50.79 | 51.25 | 51.14 |
| ID | 86.00 | 86.12 | 86.13 | 86.14 | 86.12 |

(a) DomainNet hyper-parameter ($\mu$) sweep.

**5) Visualization of the Variation in Regularization Strength across Layers over Epochs.** We present a visualization of the variation in regularization strength ($\lambda$) across different layers over epochs in Fig. 5. The results show that the regularization strength evolves dynamically during training, starting small, increasing over iterations, and eventually converging. In the vision layers (blue), early layers tend to experience weaker regularization compared to later layers throughout the training process. Conversely, the language layers (orange) display a more uniform regularization strength, with comparable levels observed between early and later layers. The weaker regularization in early vision layers likely allows them to preserve foundational low-level features, while stronger regularization in later layers encourages the model to focus on high-level semantic representations.

## 7.2 TRAINING DETAILS

**Image Classification.** For *DiGraP*, we fine-tune the model using SGD with a learning rate of $1e-2$ and $\mu = 0.1$ with a batchsize of 256. The regularization hyper-parameter is found through cross-validation, and the model with the best ID validation accuracy is taken. We use 4 RTX 2080 GPUs for each experiment.

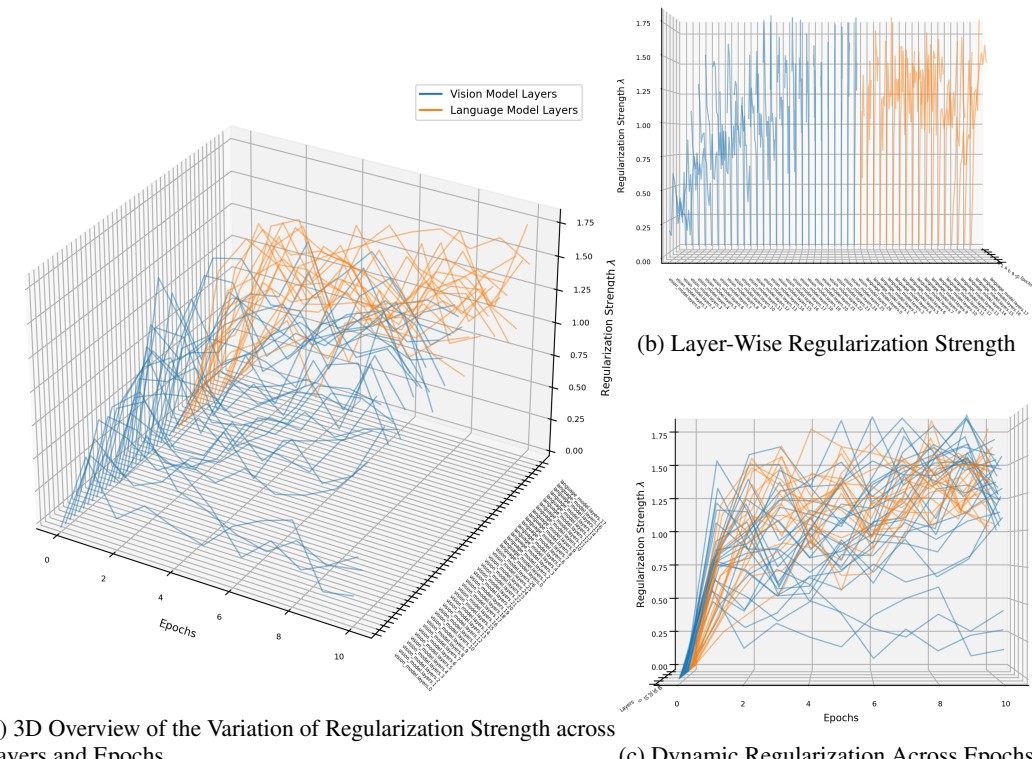

(b) Layer-Wise Regularization Strength

(a) 3D Overview of the Variation of Regularization Strength across Layers and Epochs.

(c) Dynamic Regularization Across Epochs

Figure 5: **Visualization of the Variation in Regularization Strength ($\lambda$) across Layers over Epochs.** Fig. 5a is an 3D view of R=regularization strength dynamics across layers and epochs. The X-axis represents training epochs, the Y-axis represents model layers (vision in blue, language in orange), and the Z-axis represents the regularization strength $\lambda$ applied to each layer. Fig. 5b projects the data onto the plane formed by layers (Y) and regularization strength (Z). Fig. 5c projects the data onto the plane formed by epochs (X) and regularization strength (Z).

**Fine-Tuning DomainNet-oVQA.** We use the model pretrained with $224 * 224$ input images and 128 token input/output text sequences and fine-tune with the precision of bfloat16. We use the LAVIS (Li et al., 2022) public repository to fine-tune all methods. Standard hyper-parameters are used for all: learning rate ($1e-3$), weight-decay ($1e-4$), optimizer (AdamW), scheduler (Linear Warmup With Cosine Annealing), warm-up learning rate ($1e-4$), minimum learning rate ($1e-4$), accumulation steps (2), beam size (5). The model is trained for 10 epochs with a batch size of 128 for Tab. 2. For LoRA (Hu et al., 2021), we limit our study to only adapting the attention weights and freeze the MLP modules for parameter-efficiency, specifically apply LoRA to $W_q, W_k, W_v, W_o$ with $r = 8$ in Tab. 2. We use $\lambda = 0.5$ for all *DiGraP* results in Tab. 2. The regularization hyper-parameter is found through cross-validation, and the model with the best ID validation accuracy is taken. We use 8 A40 GPU for each experiment.

**Fine-tuning VQA.** We use the model pretrained with $224 * 224$ input images and 128 token input/output text sequences and fine-tune with the precision of bfloat16. We use the LAVIS (Li et al., 2022) public repository to fine-tune all methods. Standard hyper-parameters are used for all: learning rate ($1e-3$), weight-decay ($1e-4$), optimizer (AdamW), scheduler (Linear Warmup With Cosine Annealing), warm-up learning rate ($1e-4$), minimum learning rate ($1e-4$), accumulation steps (2), beam size (5). The model is trained for 10 epochs with a batch size of 128 for Tab. 3. For LoRA (Hu et al., 2021), we limit our study to only adapting the attention weights and freeze the MLP modules for parameter-efficiency, specifically apply LoRA to $W_q, W_k, W_v, W_o$ with $r = 8$ in Tab. 3. We use $\lambda = 0.5$ for all *DiGraP* results in Tab. 3. The regularization hyper-parameter is found through cross-validation, and the model with the best ID validation accuracy is taken. We use 8 A40 GPU for each experiment.

## 7.3 MEASURING OOD DISTANCE

We follow procedures similar to typical feature-based OOD detection methods (Shi & Lee, 2024). Specifically, given our input training split $X_{\text{in}}^{\text{train}}$, we compute feature representations $z$ of the training samples to estimate the empirical mean $\mu$ and covariance matrix $\Sigma$. For each test split, we compute the test set shift relative to the training domain using the Mahalanobis distance metric defined in Eq. 9. The overall shift score for each test dataset, denoted as $S_{\text{maha}}$, is calculated as the average $S_{\text{Maha}}$ across all samples. Let $q$ denote the question, $v$ the image (vision input), and $a$ the answer. The input features used in measuring shifts include uni-modal embeddings $f(v)$, $f(q)$ and joint embeddings $f(q, v)$.

$$S_{\text{Maha}}(z_{\text{test}}) = \sqrt{(z_{\text{test}} - \mu)^\top \Sigma^{-1}(z_{\text{test}} - \mu)} \tag{9}$$

We utilize the vanilla fine-tuned PaliGemma model on the VQAv2 training dataset as our feature encoder. For the image embedding $f(v)$, we obtain it via masking out the question input tokens and mean-pooling the image portion from the final layer of the model before the language model head. Similarly, to get $f(q)$, we mask out the image tokens and extract the question portion from the final layer. To obtain $f(v, q)$, we pass in both image and text tokens as input, compute the average embedding for both modalities and then taking the overall mean.

### 7.3.1 CORRELATION BETWEEN UNI- & MULTI-MODAL SHIFTS PER DATASET

Fig. 6 shows the heatmap of the correlation between uni-modal and multi-modal shifts per dataset. Question-joint shift correlations are higher than image-joint shift correlations across all VQA datasets and fine-tuning methods. However, pre-train model maintains similar correlation between both modalities. Vanilla FT and SPD exhibits the lowest question-joint shift correlation shown by the darkest row color across all fine-tuning methods in 6a. Whilst, SPD shows the lowest image-joint shift correlation across the datasets in 6b.

### 7.3.2 HISTOGRAMS FOR EVALUATING DIFFERENT DISTRIBUTION SHIFTS

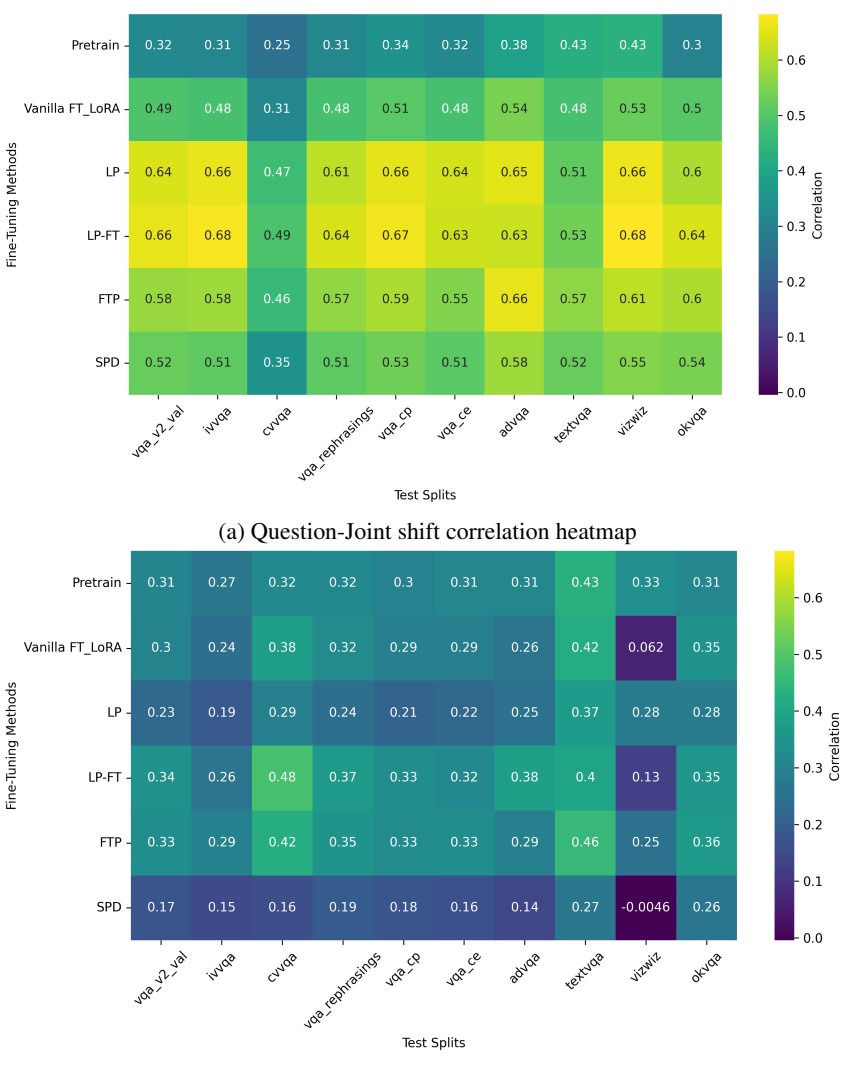

(a) Question-Joint shift correlation heatmap

(b) Image-Joint shift correlation heatmap

Figure 6: Heatmap of correlation between uni-modal and multi-modal shifts per dataset.

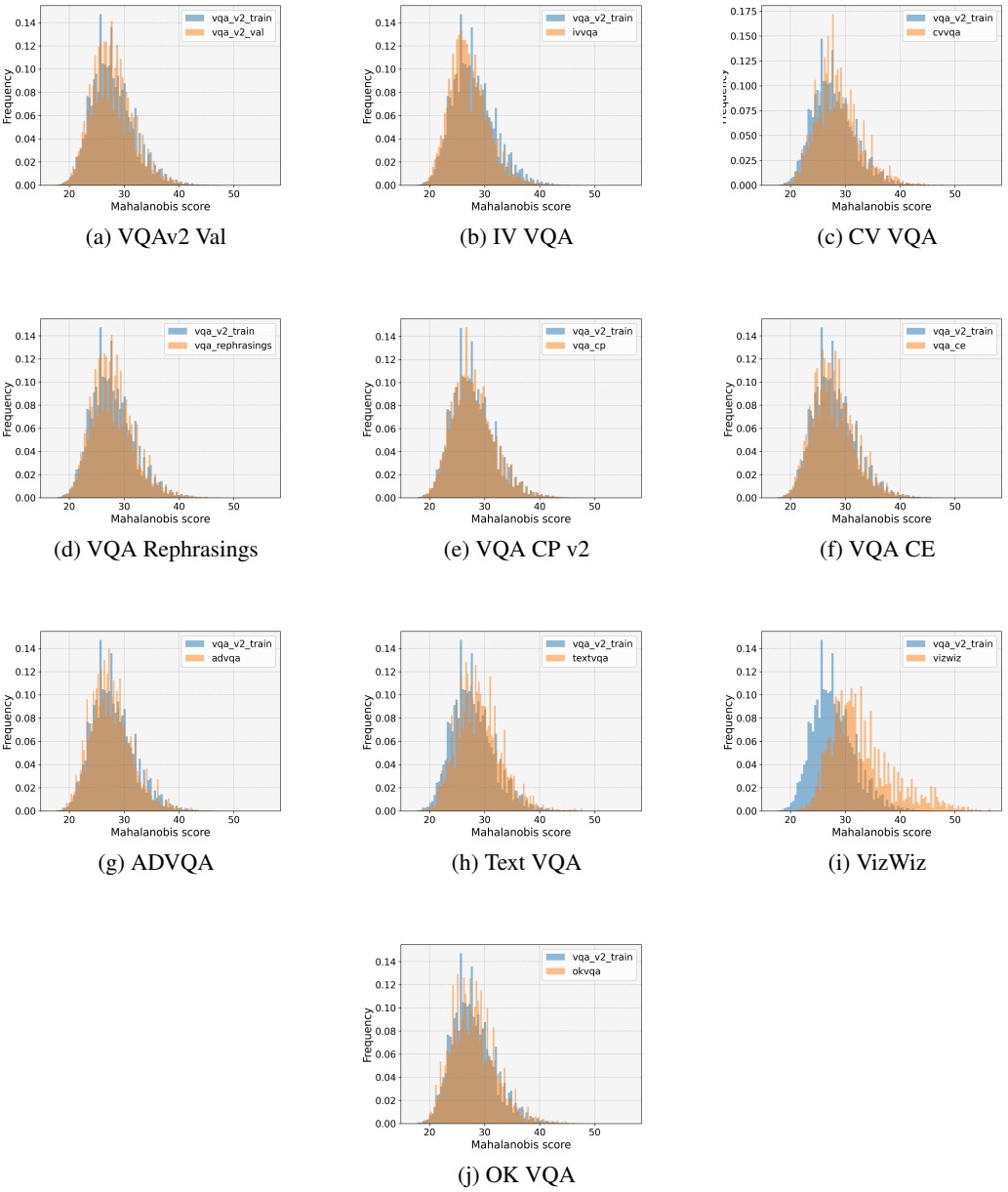

Figure 7: Histogram for Vanilla FT Visual Shifts: We depict the $S_{\text{Maha}}$ score on the visual modality for each sample in the VQAv2 train split in blue and the corresponding test samples in orange. There's minimal visual shifts for all VQA datasets from the VQAv2 train, except for Figure i which shows evidence of greater shifts between the orange distribution and the blue distribution.

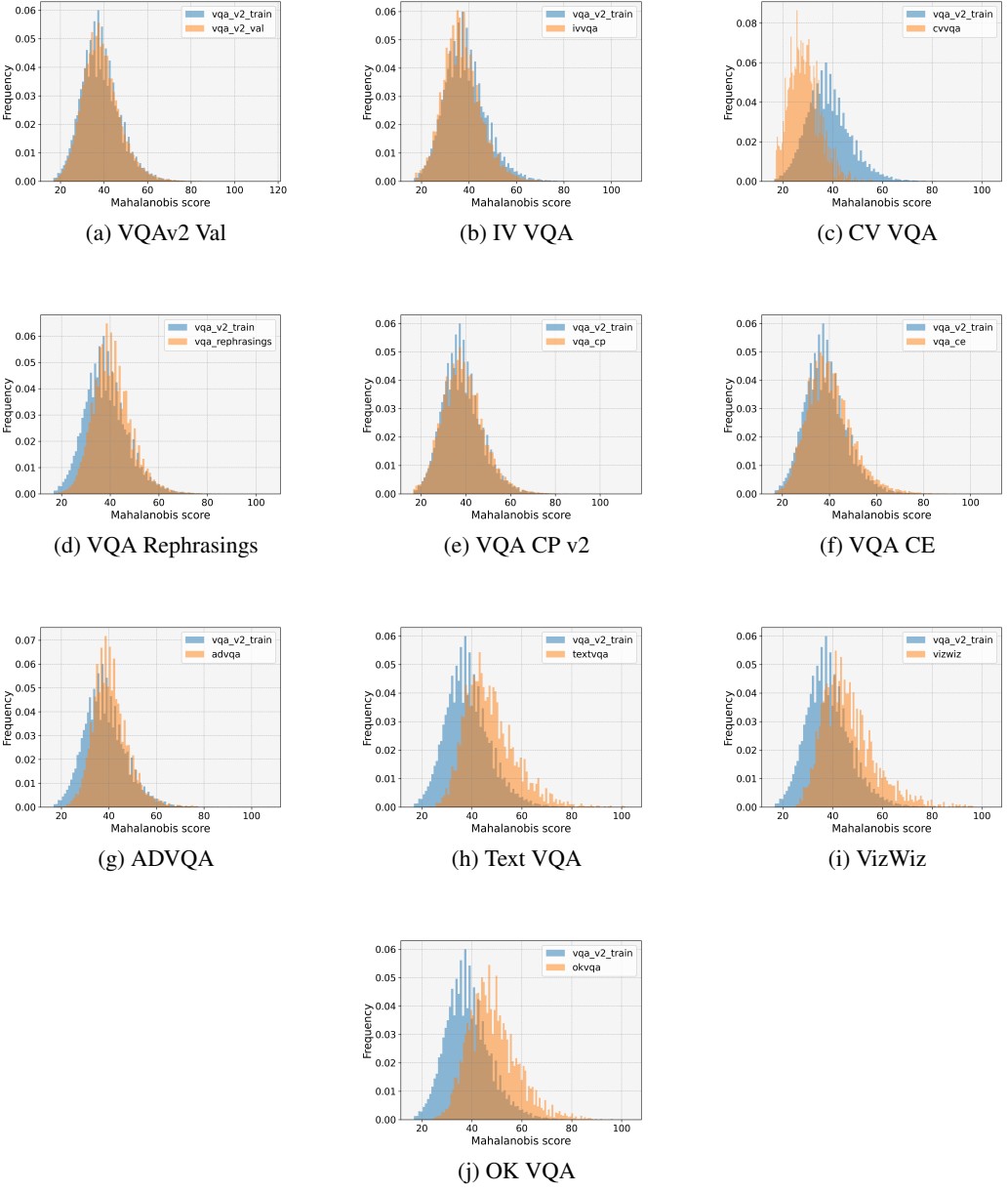

Figure 8: Histogram for Vanilla FT Question Shifts: We depict the $S_{\mathrm{Maha}}$ score on the question modality for each sample in the VQAv2 train split in blue and the corresponding test samples in orange. Similar to the visual shift histograms, far OODs (Figures h, i, j) also show evidence of greater shifts between the orange distribution and the blue distribution than near OODs.

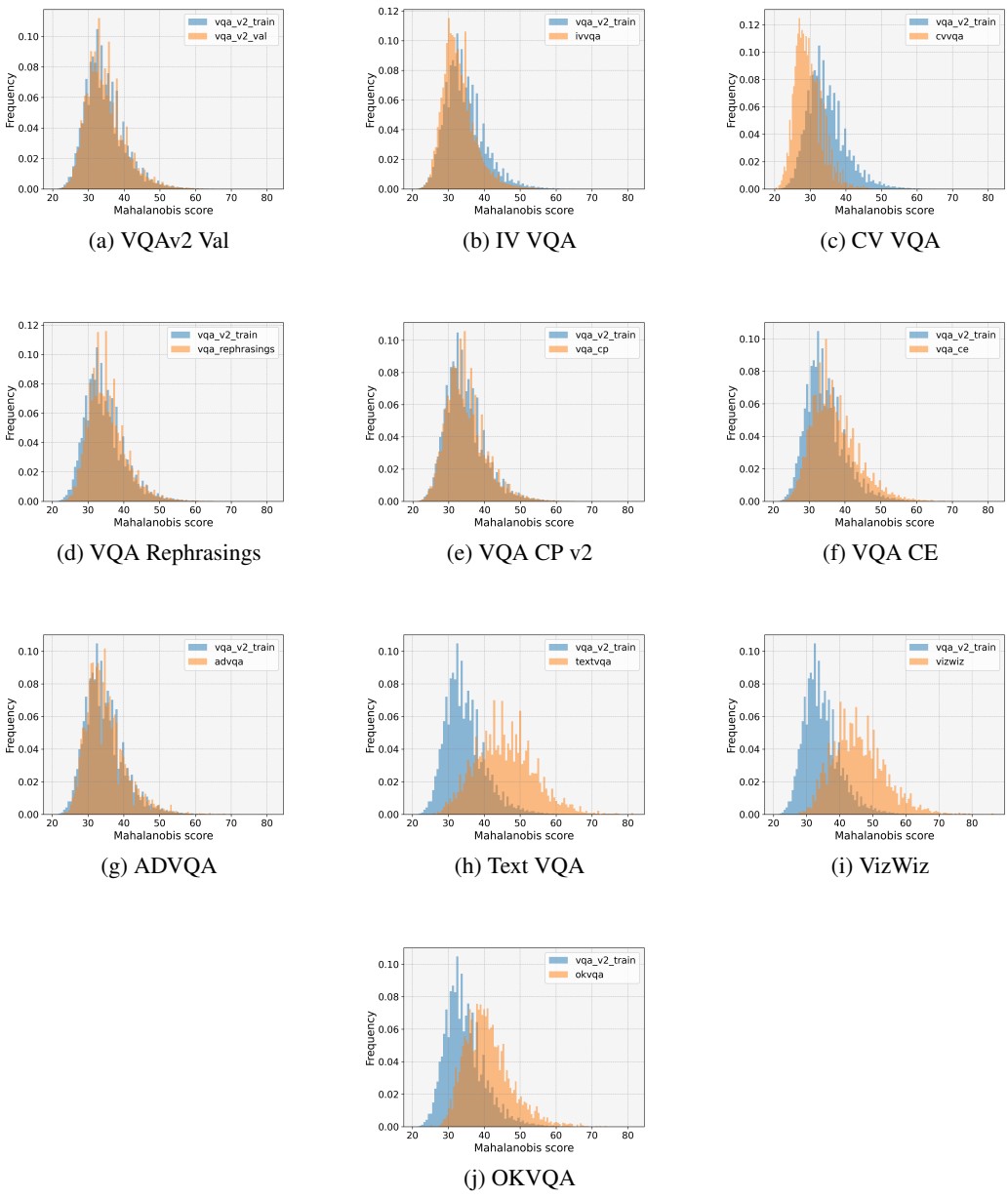

Figure 9: Histogram for Vanilla FT V+Q Shifts : We depict the $S_{\text{Maha}}$ score on the V+Q shift for each sample in the VQAv2 train split in blue and the corresponding test samples in orange. For all test splits, V+Q shifts show a greater degree of shift compared to the corresponding visual and question shift.

