# OpenReview forum: "Directional Gradient Projection for Robust Fine-Tuning of Foundation Models"
_ICLR.cc/2025/Conference — ICLR 2025 Poster_

### Official Review · Reviewer_D9Ep · 2024-11-03

**Soundness:** 3
**Presentation:** 3
**Contribution:** 3
**Rating:** 6
**Confidence:** 4

**Summary:**

The paper presents the Directional Gradient Projection (DiGraP) method, designed for robust fine-tuning of foundation models, particularly in the face of distribution shifts. DiGraP introduces a novel approach that leverages gradient directionality to resolve conflicts between optimization objectives, aiming to enhance model robustness across in-distribution (ID) and out-of-distribution (OOD) data. Unlike traditional approaches that primarily rely on weight projection and regularization, DiGraP utilizes directional information for gradient adjustments, bridging multi-objective optimization and regularization. The method’s effectiveness is demonstrated across image classification and multi-modal tasks (Visual Question Answering - VQA), outperforming baseline methods in both ID and OOD performance.

**Strengths:**

DiGraP’s gradient-based approach is innovative, incorporating directional information to handle conflicting objectives in a way that traditional regularization does not.

The expansion from uni-modal to multi-modal evaluation settings represents the contribution of robust fine-tuning.

Comprehensive ablation studies and sensitivity analyses of the hyper-parameters (e.g., projection strength) strengthen the validity of the findings, showing DiGraP’s robustness and effectiveness across different configurations.

The paper clearly explains the design choices of DiGraP, with detailed descriptions of its components.

Visualizations and tables effectively illustrate the experimental results and the model’s behavior under varying projection strengths.

**Weaknesses:**

There are many public unimodal and multimodal foundation models, e.g., MAE, CLIP, BEiT3, LLaVa, etc. It is unclear why ResNet50 and PaliGemma are selected as foundation models. The ResNet50 pretrained in a supervised manner on ImageNet can hardly be deemed as foundation models. The PaliGemma is pretrained on a broad mixture of large-scale vision-language tasks. Whether the conclusion of this paper holds across other, more general foundation models, e.g., CLIP or LLaVa, is questionable.

Although DiGraP demonstrates improved performance in near OOD settings, its performance on far OOD tasks remains limited compared to other methods. The authors acknowledge this trade-off between ID/near OOD and far OOD robustness, but a deeper investigation into addressing this limitation would enhance the model’s versatility.

The model’s gradient projection mechanism, while theoretically sound, lacks interpretability in how projection strength decisions impact specific instances.

The figures in Appendix have too small font.

**Questions:**

Could additional techniques be incorporated to enhance DiGraP’s performance on far OOD datasets without sacrificing ID and near OOD performance?

Could the authors provide more insights into how the projection strength parameter adapts dynamically across different layers and tasks, especially in multi-modal settings?

---

> ### Author Response · Authors · 2024-11-24
>
> We appreciate the reviewer’s kind comments and the suggestion of adding more backbones and visualizations of the projection strength across the layers during training. We have included thorough answers in our response and are happy to engage further during the rebuttal period if clarification is needed.
>
> **There are many public unimodal and multimodal foundation models, e.g., MAE, CLIP, BEiT3, LLaVA, etc. It is unclear why ResNet50 and PaliGemma are selected as foundation models. The ResNet50 pretrained in a supervised manner on ImageNet can hardly be deemed as foundation models. The PaliGemma is pretrained on a broad mixture of large-scale vision-language tasks. Whether the conclusion of this paper holds across other, more general foundation models, e.g., CLIP or LLaVA, is questionable.**
>
> We adopt the MOCO-V3 ResNet50 as the backbone for the image classification task, following previous work [1] [2]. MOCO-V3 ResNet50 builds on the ResNet50 architecture but is pre-trained on ImageNet-1k in a self-supervised manner rather than supervised. For the multi-modal backbone, we use PaliGemma-3B, a recently released lightweight model by Google that achieves state-of-the-art performance on VQAv2.
>
> For this rebuttal, we additionally include results for DomainNet with CLIP ViT-Base (Tab. 10), DomainNet-oVQA (Tab. 9), and VQA (Tab. 8) with LLaVA in the **Additional Experiments for Rebuttal Section** in Appendix. DiGraP consistently achieves the best performance across all experiments. Thank you for this suggestion, and we will include these results in the paper.
>
> ---
>
> **Although DiGraP demonstrates improved performance in near OOD settings, its performance on far OOD tasks remains limited compared to other methods. The authors acknowledge this trade-off between ID/near OOD and far OOD robustness, but a deeper investigation into addressing this limitation would enhance the model’s versatility.**
> **Could additional techniques be incorporated to enhance DiGraP’s performance on far OOD datasets without sacrificing ID and near OOD performance?**
>
> Thanks for pointing this out. Enhancing robustness across both near and far OOD settings is a challenging and underexplored problem. Most prior work, such as [1] [2] [3] [4], focuses on general OOD robustness without explicitly differentiating between near and far OOD scenarios. Addressing DiGraP’s limitations on far OOD datasets while maintaining strong ID and near OOD performance requires a more detailed investigation into the dynamics of projection strength and its interaction with diverse distribution shifts. This remains a promising direction for future research, which could significantly enhance the versatility and robustness of DiGraP across a wider range of tasks.
>
> ---
>
> **Could the authors provide more insights into how the projection strength parameter adapts dynamically across different layers and tasks, especially in multi-modal settings?**
>
> We present a visualization of the variation in regularization strength ($\lambda$) across different layers over epochs in Fig. 9 of the **Additional Experiments for Rebuttal Section** in Appendix. The results show that the regularization strength evolves dynamically during training, starting small, increasing over iterations, and eventually converging. In the vision layers (blue), early layers tend to experience weaker regularization compared to later layers throughout the training process. Conversely, the language layers (orange) display a more uniform regularization strength, with comparable levels observed between early and later layers.
>
> The weaker regularization in early vision layers likely allows them to preserve foundational low-level features, while stronger regularization in later layers encourages the model to focus on high-level semantic representations. Thank you for the suggestion; we will include these visualizations in the paper, hoping they inspire additional ideas for future work.

---

> > ### Author Response · Authors · 2024-11-24
> >
> > ---
> > **The model’s gradient projection mechanism, while theoretically sound, lacks interpretability in how projection strength decisions impact specific instances.**
> >
> > DiGraP is designed to balance pre-trained and fine-tuned trajectories by dynamically adjusting projection strength to optimize overall training directions. This adjustment can be interpreted layer-wise, showing how different layers contribute to balancing pre-trained knowledge and fine-tuning, and iteration-wise, revealing how projection strength evolves during training. However, it lacks interpretability at the instance level, as it focuses on global optimization across layers and iterations rather than tailoring projection strength to individual data instances.
> >
> > ---
> >
> > **The figures in Appendix have too small font.**
> >
> > Thanks for the suggestion! We will increase the font size and update the figures.
> >
> >
> > [1] Junjiao Tian, Xiaoliang Dai, Chih-Yao Ma, Zecheng He, Yen-Cheng Liu, and Zsolt Kira. Trainable Projected Gradient Method for Robust Fine-tuning, March 2023a.
> >
> > [2] Junjiao Tian, Yen-Cheng Liu, James Seale Smith, and Zsolt Kira. Fast Trainable Projection for Robust Fine-Tuning, October 2023b.
> >
> > [3] Mitchell Wortsman, Gabriel Ilharco, Jong Wook Kim, Mike Li, Simon Kornblith, Rebecca Roelofs Raphael Gontijo-Lopes, Hannaneh Hajishirzi, Ali Farhadi, Hongseok Namkoong, and Ludwig Schmidt. Robust fine-tuning of zero-shot models, June 2022
> >
> > [4] Ananya Kumar, Aditi Raghunathan, Robbie Jones, Tengyu Ma, and Percy Liang. Fine-Tuning can Distort Pretrained Features and Underperform Out-of-Distribution, February 2022.

---

> ### Author Response · Authors · 2024-11-26
> **Sincerely Awaiting Your Feedback**
>
> Dear Reviewer D9Ep,
>
> I hope this message finds you well. With the rebuttal deadline fast approaching, we kindly seek your additional feedback to help us address any remaining concerns and ensure the highest quality of this work.
>
> Your insights and suggestions would be immensely valuable in guiding our revision process and strengthening our submission. We deeply appreciate your time and expertise and look forward to your response.
>
> Thank you once again for your thoughtful review.
>
> Best regards,
> Authors

---

> ### Author Response · Authors · 2024-12-02
>
> Dear Reviewer D9Ep,
>
> Sorry for the confusion. The Additional Experiments for Rebuttal Section in Appendix is in the 19-20 pages in the main PDF. We greatly appreciate your detailed feedback, which has been invaluable in improving our work. Since the discussion deadline is approaching and the rating is still in borderline state, we would like to kindly request if you could provide any additional comments or clarifications that might help us address your concerns further. Your input is crucial in helping us improve the contribution of our work to the community.
>
> Please let us know if there is anything more we can clarify or elaborate on to assist in your evaluation. Thank you again for your efforts and valuable time.
>
> Best regards,
> Authors

---

> ### Author Response · Authors · 2024-12-02
>
> Dear Reviewer D9Ep,
>
> We sincerely appreciate your thoughtful response and the support you have shown for our work. Your specific suggestions for additional analyses have been incredibly helpful in enhancing the quality of this manuscript.
>
> Thank you once again for your valuable feedback and encouragement.
>
> Best regards,
> The Authors

---

### Official Review · Reviewer_1rWn · 2024-11-03

**Soundness:** 3
**Presentation:** 3
**Contribution:** 2
**Rating:** 5
**Confidence:** 3

**Summary:**

This paper proposes a novel layer-wise trainable method for robust fine-tuning of foundation models, namely DiGraP, which considers the directional information of the gradients. In addition, the authors generalize the method to multi-modal evaluation settings, filling the gap in experiment. Finally, experiments on several tasks demonstrate the effectiveness and robustness of the proposed method.

**Strengths:**

+This work adopting multi-objective learning to alleviate the previous method's sensitivity to hyper-parameters and underfitting problems, which seems to a relatively novel attempt in the current field.

+The work expands the experimental setting from single-modal to multi-modal, filling the gap in experiments.

**Weaknesses:**

-Although the work attempts to utilize gradient direction information, there are still deficiencies in the innovation of the method. It looks like a combination of previous work, so the innovation needs to be explained more.

-Sec 4.1 lacks qualitative analysis of the experimental results, especially the reasons why the ID performance on the real domain is worse than LP-FT.

-The article seems to have only conducted a quantitative analysis of hyper-parameters for multi-modal tasks, and corresponding ablation experiments are still needed on singal-modal tasks.

**Questions:**

Please refer to the weaknesses.

---

> ### Author Response · Authors · 2024-11-24
>
> Thank you for your positive remarks and for suggesting additional qualitative studies and hyperparameter tuning experiments. We have provided comprehensive answers in our response and would be happy to discuss further during the rebuttal period if required.
>
> **Although the work attempts to utilize gradient direction information, there are still deficiencies in the innovation of the method. It looks like a combination of previous work, so the innovation needs to be explained more.**
>
> The distinctions between DiGraP and PCGrad can be summarized as follows:
> 1. **Purpose**: PCGrad is developed for addressing general multi-objective optimization challenges, whereas DiGraP is designed specifically to enhance robust fine-tuning. DiGraP introduces a novel perspective by framing robust fine-tuning as a multi-objective optimization problem, setting it apart from conventional methods that rely on regularization techniques or bi-level optimization frameworks.
> 2. **Flexibility and Automation**: PCGrad requires users to manually assign weights to different objectives and fixes the projection strength as a static hyperparameter throughout the training process. In contrast, DiGraP leverages a hyper-optimizer to automatically learn the relative importance of objectives, allowing the projection strength to dynamically adjust at different layers and adapt as training progresses. This capability makes DiGraP significantly more flexible and robust than manually tuned approaches and is hence a crucial enhancement to the specific problem that we tackle (robust fine-tuning).
>
> ---
>
> **Sec 4.1 lacks qualitative analysis of the experimental results, especially the reasons why the ID performance on the real domain is worse than LP-FT.**
>
> We note that DiGraP performs better on ID than vanilla fine-tuning in Table 1 and outperforms all other methods in all other tables. Additionally, LP-FT's improved ID performance comes at the greater cost of OOD performance. For the rebuttal, we equalized LP-FT's ID performance as in TPGM-C from [1], where we performed an LP-FT-C experiment (a controlled variant of LP-FT) in Tab. 6 by increasing the regularization strength to align the ID performance with that of DiGraP. Even with this adjustment, DiGraP demonstrates superior performance compared to LP-FT-C on OOD datasets.
>
> ---
>
> **The article seems to have only conducted a quantitative analysis of hyperparameters for multi-modal tasks, and corresponding ablation experiments are still needed on single-modal tasks.**
>
> We include the hyperparameter tuning experiment for DomainNet in Tab. 7 of the Additional Experiments for Rebuttal Section in Appendix. The results demonstrate that DiGraP remains robust to hyperparameter variations on both ID and OOD datasets in the uni-modal image classification task.
>
> [1] Junjiao Tian, Xiaoliang Dai, Chih-Yao Ma, Zecheng He, Yen-Cheng Liu, and Zsolt Kira. Trainable
> Projected Gradient Method for Robust Fine-tuning, March 2023a.

---

> ### Author Response · Authors · 2024-11-26
> **Sincerely Awaiting Your Feedback**
>
> Dear Reviewer 1rWn,
>
> I hope this message finds you well. With the rebuttal deadline fast approaching, we kindly seek your additional feedback to help us address any remaining concerns and ensure the highest quality of this work.
>
> Your insights and suggestions would be immensely valuable in guiding our revision process and strengthening our submission. We deeply appreciate your time and expertise and look forward to your response.
>
> Thank you once again for your thoughtful review.
>
> Best regards,
> Authors

---

> ### Author Response · Authors · 2024-12-04
>
> Dear Reviewer 1rWn,
>
> We greatly appreciate your detailed feedback, which has been invaluable in improving our work. Since the discussion deadline is approaching and the rating is still in borderline state, we would like to kindly request if you could provide any additional comments or clarifications that might help us address your concerns further. Your input is crucial in helping us improve the contribution of our work to the community.
>
> Please let us know if there is anything more we can clarify or elaborate on to assist in your evaluation. Thank you again for your efforts and valuable time.
>
> Best regards, Authors

---

### Official Review · Reviewer_fvUQ · 2024-11-09

**Soundness:** 3
**Presentation:** 4
**Contribution:** 2
**Rating:** 6
**Confidence:** 4

**Summary:**

The paper proposes a new method for robust fine-tuning of models for robust fine-tuning that maintains model performance on out-of-distribution datasets while fine-tuning on in-domain data. The authors frame robust finetuning as a multi-objective optimization problem where the two objectives are the original loss function and the distance between the fine-tuned and pretrained weights. They propose an algorithm which projects the gradient of the original loss onto the normal plane of the regularization term’s gradient when the two gradients do not conflict. This approach is evaluated on image classification and VQA tasks, demonstrating improved performance on both in-domain and OOD datasets.

**Strengths:**

1. The proposed approach makes the projection strength $\omega$ as a learnable parameter, and makes it tunable by a learning rate $\mu$. This makes the approach less sensitive to hyperparameters. The authors show good experimental evidence for the claim.
2. The proposed evaluation of reformulating image classification as a VQA task allows for evaluations on foundation VLMs like PaliGemma.
3. The authors show that the proposed approach can be adapted to PEFT approaches like LoRA.

**Weaknesses:**

1. While the paper addresses the problem of robust fine-tuning, the proposed approach Directional Gradient Projection appears similar to PCGrad. The problem of conflicting gradients and the  idea of projecting the gradient to the normal plane have been addressed in PCGrad. I would encourage the authors to highlight the unique contributions more clearly.
2. The paper proposes a general approach for robust fine-tuning of foundation models, but the paper focuses on Image classification and VQA tasks. It would be useful to evaluate the approach by fine-tuning open-source LLMs like LLaMA on a multi-task benchmark like MMLU.

**Questions:**

1. In Table 1, while DiGraP performs well on OOD domains, the performance on the ID domain is poor compared to other approaches like LP-FT and TPGM. By changing the strength of ‘w’, does the approach allow a trade-off to improve performance on ID while compromising OOD performance?
2. In Table 4(b), when the learning rate is increased, which makes the projection strength larger, why does the ID performance improve?

---

> ### Author Response · Authors · 2024-11-24
>
> We thank the reviewer for the positive feedback and for suggesting an ablation study on the ID and OOD trade-off, as well as insightful new tasks. We have provided detailed answers in our response and are happy to discuss further during the rebuttal period if needed.
>
> **While the paper addresses the problem of robust fine-tuning, the proposed approach Directional Gradient Projection appears similar to PCGrad. The problem of conflicting gradients and the idea of projecting the gradient to the normal plane have been addressed in PCGrad. I would encourage the authors to highlight the unique contributions more clearly.**
>
> The differences between DiGraP and PCGrad are as follows:
> 1. **Objective Focus**: PCGrad is designed to address multi-objective optimization problems, while DiGraP is specifically tailored for robust fine-tuning. Unlike previous robust fine-tuning approaches that treat it as a regularization task or a bi-level optimization problem, DiGraP is the first to frame robust fine-tuning as a multi-objective optimization problem.
> 2. **Weighting and Adaptability**: PCGrad requires manually defining weights for different objectives and treats projection strength as a fixed hyperparameter throughout training. In contrast, DiGraP dynamically learns the dominance of different objectives through a hyper-optimizer, allowing the projection strength to adapt across layers and evolve during training. This adaptive approach is significantly more robust compared to manually defined hyperparameters and is hence a crucial enhancement to the specific problem that we tackle (robust fine-tuning).
>
> ---
>
> **The paper proposes a general approach for robust fine-tuning of foundation models, but the paper focuses on image classification and VQA tasks. It would be useful to evaluate the approach by fine-tuning open-source LLMs like LLaMA on a multi-task benchmark like MMLU.**
>
> Thank you for pointing this out. Previous work on robust fine-tuning has primarily focused on image classification, with limited benchmarks addressing language distribution shifts. In this paper, we propose *a comprehensive multi-modal setting*, which includes language and language-specific shifts such as VQA-Rephrasings (question shift) and VQA-CP (answer shift). While MMLU is a multi-task benchmark, it does not include settings related to distribution shifts, which differs from the focus of our work.
>
> Nevertheless, we provide additional experiments on different backbones to further demonstrate the effectiveness of DiGraP. The results of these experiments are in the **Additional Experiments for Rebuttal Section** in Appendix:
> - DomainNet with CLIP ViT-Base (Tab. 10),
> - DomainNet-oVQA with LLaVA (Tab. 9),
> - VQA with LLaVA (Tab. 8).
>
> Currently, we are working on in-distribution experiments on MMLU and will share the results once they are complete.
>
> ---
>
> **In Table 1, while DiGraP performs well on OOD domains, the performance on the ID domain is poor compared to other approaches like LP-FT and TPGM. By changing the strength of ‘w,’ does the approach allow a trade-off to improve performance on ID while compromising OOD performance?**
>
> We note that DiGraP outperforms vanilla fine-tuning on ID in Table 1 and surpasses all other methods in all other tables. Additionally, LP-FT's improved ID performance comes at the cost of significantly lower OOD performance. For the rebuttal, we conducted an LP-FT-C experiment (a controlled version of LP-FT), similar to TPGM-C in [1], by increasing the regularization strength to match the ID performance of DiGraP (Tab. 6). Despite this adjustment, DiGraP still outperforms LP-FT-C on OOD datasets.
>
> ---
>
> **In Table 4(b), when the learning rate is increased, which makes the projection strength larger, why does the ID performance improve?**
>
> When the learning rate is increased, the overall projection strength becomes larger. However, the individual projection strength for each layer adjusts dynamically during training. This allows the model to simultaneously preserve pre-trained robustness and adapt to downstream tasks, preventing a decrease in ID performance. This dynamic adjustment is a key advantage of our adaptive approach to robust fine-tuning.
>
> [1] Junjiao Tian, Xiaoliang Dai, Chih-Yao Ma, Zecheng He, Yen-Cheng Liu, and Zsolt Kira. Trainable
> Projected Gradient Method for Robust Fine-tuning, March 2023a.

---

> ### Author Response · Authors · 2024-11-26
> **Sincerely Awaiting Your Feedback**
>
> Dear Reviewer fvUQ,
>
> I hope this message finds you well. With the rebuttal deadline fast approaching, we kindly seek your additional feedback to help us address any remaining concerns and ensure the highest quality of this work.
>
> Your insights and suggestions would be immensely valuable in guiding our revision process and strengthening our submission. We deeply appreciate your time and expertise and look forward to your response.
>
> Thank you once again for your thoughtful review.
>
> Best regards,
> Authors

---

> > ### Comment · Reviewer_fvUQ · 2024-11-27
> > **Thank for for addressing my concerns**
> >
> > Dear Authors,
> >
> > Thank you for addressing my concerns and performing additional experiments to strengthen the paper.
> >
> > Framing robust fine-tuning as a multi-objective optimization problem and introducing the hyper-optimizer indeed makes DiGraP a novel and valuable contribution to the field of robust fine-tuning. However, I would like to highlight that the concepts of conflicting gradients and projecting gradients onto the normal plane were initially proposed in PCGrad. From Figure 1, it appears that these ideas are presented as novel contributions of DiGraP. I kindly request the authors to acknowledge the origins of these ideas in the background section and mention contributions that unique to DiGraP in the final manuscript.
> >
> > I also appreciate the inclusion of experiments with other VLMs as the base model. This provides further validation of the proposed approach and strengthens the paper.
> >
> > While I am inclined toward the paper being accepted and will maintain my original score, I look forward to the responses and discussions from other reviewers before making my final recommendation.

---

> > > ### Author Response · Authors · 2024-11-28
> > >
> > > Dear Reviewer fvUQ,
> > >
> > > Thank you for your thoughtful feedback and support for our work! We have made the following updates to address your concerns:
> > >
> > > 1. **Acknowledgment in the Background Section:**
> > >    We have updated the background in Sec. 3.2 (lines 153–158) to explicitly acknowledge the origins of conflicting gradient concepts in PCGrad.
> > >
> > > 2. **Clarification of Unique Contributions:**
> > >    Our unique contributions are now clarified in Sec. 3.3 (lines 249–253) and highlighted in blue to distinguish them from prior work.
> > >
> > > 3. **Additional Experiment:**
> > >    We have added a new experiment in Table 11 using BOSS [1], an NLP benchmark suite designed for OOD robustness evaluation. This suite includes both ID and OOD language-only datasets across multiple tasks (e.g., Sentiment Analysis, Toxic Detection). We hope this experiment could further address your concern regarding the broader applicability of our method.
> > >
> > > Thank you once again for your invaluable feedback and encouragement. We hope the updates sufficiently address your comments and improve the clarity of our work.
> > >
> > > [1] Lifan Yuan, Yangyi Chen, Ganqu Cui, Hongcheng Gao, Fangyuan Zou, Xingyi Cheng, Heng Ji,
> > > Zhiyuan Liu, and Maosong Sun. Revisiting out-of-distribution robustness in nlp: Benchmark,
> > > analysis, and llms evaluations, 2023.
> > >
> > > Best regards,
> > > Authors

---

> ### Author Response · Authors · 2024-12-04
>
> Dear Reviewer fvUQ,
>
> We greatly appreciate your detailed feedback, which has been invaluable in improving our work. We have updated the manuscript accordingly. Since the discussion deadline is approaching, we kindly request if you could provide any additional comments or clarifications that might help us address your concerns further. Your input is crucial in enhancing the contribution of our work to the community.
>
> Please let us know if there is anything more we can clarify or elaborate on to assist in your evaluation. Thank you again for your efforts and valuable time.
>
> Best regards, Authors

---

### Author Response · Authors · 2024-11-24

We thank all the reviewers for their positive feedback on this work's adaptability (fvUQ), performance (fvUQ, D9Ep), and insights (fvUQ, 1rWn, D9Ep). We aim to address your questions with concrete responses and clarify any points of confusion.

The **Additional Experiments for Rebuttal in Appendix (19-20 pages in main PDF)** includes new experiments and studies requested by the reviewers. The new experiments are summarized below:

- **(D9Ep)** We add experiments of fine-tuning on DomainNet with CLIP ViT-Base, DomainNet-oVQA, and VQA with LLaVA-7B. DiGraP consistently achieves the best performance across all experiments.
- **(fvUQ, 1rWn)** We conduct an LP-FT-C experiment (a controlled version of LP-FT) by increasing the regularization strength to ensure that the ID performance matches that of DiGraP. Despite this adjustment, DiGraP still outperforms LP-FT-C on OOD datasets.
- **(1rWn)** We include the hyper-parameter tuning experiment for DomainNet on CLIP ViT-Base. The results demonstrate that DiGraP remains robust to hyperparameter variations on both ID and OOD datasets in the uni-modal image classification task.
- **(D9Ep)** We present a visualization of the variation in regularization strength (λ) across different layers over epochs.

We hope our response addresses your questions and welcome further discussion during the rebuttal period.

---

### Meta-Review · Area_Chair_ahJx · 2024-12-23

**Metareview:**

This paper introduces Directional Gradient Projection (DiGraP), a robust fine-tuning approach that utilizes gradient-based optimization to address conflicting objectives during the fine-tuning of foundation models. The reviewers collectively acknowledged the paper’s strengths and its potential contributions to the field, though they also noted certain weaknesses.

All reviewers commended the novelty of introducing a learnable projection strength parameter, which effectively reduces sensitivity to hyperparameters. Additionally, the expansion of evaluation settings from uni-modal to multi-modal tasks was widely appreciated as a significant step.

On the downside, concerns were raised regarding the novelty of DiGraP against PCGrad, the choice of models such as ResNet50, and the limited performance on far out-of-distribution (OOD) tasks. The authors addressed these concerns in their rebuttal. Reviewer fvUQ, who engaged in the discussion phase, appreciated the authors’ responses and expressed a positive inclination toward accepting the paper. Unfortunately, the other two reviewers did not participate in further discussions post-rebuttal.

The AC panel concurs with the positive sentiment of the reviewers. Despite some limitations, the proposed method is technically sound, well-articulated, and supported by promising experimental results. The strengths of the work, including its theoretical contributions and practical utility, outweigh the identified weaknesses.

The AC panel, therefore, recommends accepting the paper.

**Additional Comments On Reviewer Discussion:**

The reviewer (fvUQ) thanked the authors for addressing the concerns raised  and performing additional experiments. The reviewer expressed his inclination towards accepting the paper. Other two reviewers did not participate in the discussion and retained their.

---

### Decision · Program_Chairs · 2025-01-22

Accept (Poster)